# Riemannian Denoising Diffusion Probabilistic Models

## Abstract

We propose Riemannian Denoising Diffusion Probabilistic Models (RDDPMs) for learning distributions on submanifolds of Euclidean space that are level sets of functions, including most of the manifolds relevant to applications. Existing methods for generative modeling on manifolds rely on substantial geometric information such as geodesic curves or eigenfunctions of the Laplace-Beltrami operator and, as a result, they are limited to manifolds where such information is available. In contrast, our method, built on a projection scheme, can be applied to more general manifolds, as it only requires being able to evaluate the value and the first order derivatives of the function that defines the submanifold. We provide a theoretical analysis of our method in the continuous-time limit, which elucidates the connection between our RDDPMs and score-based generative models on manifolds. The capability of our method is demonstrated on datasets from previous studies and on new datasets sampled from two high-dimensional manifolds, i.e. $SO(10)$ and the configuration space of molecular system alanine dipeptide with fixed dihedral angle.

## 1 Introduction

Diffusion models are generative models that learn data distributions by gradually perturbing data into noise and then reconstructing data from noise using stochastic processes. Two primary frameworks of diffusion models are Denoising Diffusion Probabilistic Models (DDPMs;Ho et al. (2020)), where the models are trained to minimize the variational bound in variational inference, and Score-based Generative Models (SGMs;Song & Ermon (2019); Song et al. (2021b)), where the models are trained to learn the score function (Hyvärinen & Dayan, 2005). Both frameworks have achieved remarkable success in various applied fields.

In recent years, there has been a growing interest in developing generative models for data on manifolds (De Bortoli et al., 2022; Lou et al., 2023; Chen & Lipman, 2024; Jo & Hwang, 2024). However, existing methods on manifolds rely on substantial geometric information, e.g. geodesics (De Bortoli et al., 2022), heat kernel or its approximations (De Bortoli et al., 2022; Lou et al., 2023), or eigenfunctions and (pre)metrics (Chen & Lipman, 2024). As a result, their applications are restricted to manifolds where such information can be obtained.

In this work, we introduce Riemannian Denoising Diffusion Probabilistic Models (RDDPMs), an extension of DDPMs to Riemannian submanifolds. A key ingredient is the projection scheme used in Monte Carlo methods for sampling under constraints (Zappa et al., 2018; Lelièvre et al., 2022), which allows us to develop Markov chains on manifolds with explicit transition densities. The main advantages of our method over existing methods are summarized below.

- Our method is developed for submanifolds that are level sets of smooth functions in Euclidean space. This general setting includes most of the often studied manifolds such as spheres and matrix groups. More importantly, it fits well with applications where constraints are involved, e.g. applications in molecular dynamics and statistical mechanics.

- Our method only requires the computation of the value and the first order derivatives of the function that defines the submanifold. Therefore, it is applicable to more general manifolds.

- We present a theoretical analysis for the loss function of our method in the continuous-time limit, elucidating its connection to the existing methods (De Bortoli et al., 2022). This analysis also

shows the equivalence between loss functions derived from variational bound in variational inference and from learning score function.

We successfully apply our method to datasets from prior works, and to new datasets from the special orthogonal group SO(10) and from alanine dipeptide with fixed dihedral angle, both of which are challenging for existing methods due to their geometric complexity.

## 2 BACKGROUND

**Riemannian submanifolds** We consider the zero level set $\mathcal{M} = \{x \in \mathbb{R}^n | \xi(x) = 0\}$ of a smooth function $\xi : \mathbb{R}^n \to \mathbb{R}^{n-d}$. We assume that $\mathcal{M}$ is non-empty and the matrix $\nabla\xi(x) \in \mathbb{R}^{n \times (n-d)}$, i.e. the Jacobian of $\xi$, has full rank at each $x \in \mathcal{M}$. Under this assumption, $\mathcal{M}$ is a $d$-dimensional submanifold of $\mathbb{R}^n$. We further assume that $\mathcal{M}$ is a smooth compact connected manifold without boundary. The Riemannian metric on $\mathcal{M}$ is endowed from the standard Euclidean distance on $\mathbb{R}^n$. For $x \in \mathcal{M}$, we denote by $T_x\mathcal{M}$ the tangent space of $\mathcal{M}$ at $x$. The orthogonal projection matrix $P(x) \in \mathbb{R}^{n \times n}$ mapping $T_x\mathbb{R}^n = \mathbb{R}^n$ to $T_x\mathcal{M}$ is given by $P(x) = I_n - \nabla\xi(x)\big(\nabla\xi(x)^\top \nabla\xi(x)\big)^{-1}\nabla\xi(x)^\top$. Let $U_x \in \mathbb{R}^{n \times d}$ be a matrix whose column vectors form an orthonormal basis of $T_x\mathcal{M}$ such that $U_x^\top U_x = I_d$. It is straightforward to verify that $P(x) = U_x U_x^\top$. The volume element over $\mathcal{M}$ is denoted by $\sigma_\mathcal{M}$. All probability densities that appear in this paper refer to relative probability densities with respect to either $\sigma_\mathcal{M}$ or the product of $\sigma_\mathcal{M}$ over product spaces. For notational simplicity, we also use the shorthand $\int p(x^{(1:N)})\, dx^{(1:N)} := \int_\mathcal{M} \cdots \int_\mathcal{M} p(x^{(1)}, x^{(2)}, \ldots, x^{(N)})\, d\sigma_\mathcal{M}(x^{(1)})\, d\sigma_\mathcal{M}(x^{(2)}) \cdots d\sigma_\mathcal{M}(x^{(N)})$.

**Denoising diffusion probabilistic models** We formulate the DDPMs (Sohl-Dickstein et al., 2015; Ho et al., 2020) to the general Riemannian manifold setting. Assume that the data distribution is $q_0(x)d\sigma_\mathcal{M}(x)$. DDPMs are a class of generative models built on Markov chains. Specifically, states $x^{(1)}, \ldots, x^{(N)} \in \mathcal{M}$ are generated by gradually corrupting the data $x^{(0)}$ according to a Markov chain on $\mathcal{M}$, i.e. the forward process. The joint probability density of $x^{(1)}, \ldots, x^{(N)}$ given $x^{(0)}$ is

$$q(x^{(1:N)} \,|\, x^{(0)}) = \prod_{k=0}^{N-1} q(x^{(k+1)} \,|\, x^{(k)}). \tag{1}$$

The generative process, also called the reverse process, is a Markov chain on $\mathcal{M}$ that is learnt to reproduce the data by reversing the forward process. Its joint probability density is

$$p_\theta(x^{(0:N)}) = p(x^{(N)}) \prod_{k=0}^{N-1} p_\theta(x^{(k)} \,|\, x^{(k+1)}), \tag{2}$$

where $p(x^{(N)})$ is a (fixed) prior density. The probability density of $x^{(0)}$ generated by the reverse process is therefore $p_\theta(x^{(0)}) = \int p_\theta(x^{(0:N)})\, dx^{(1:N)}$. The learning objective is based on the standard variational bound on the negative log-likelihood. Specifically, using equations 1–2, and applying Jensen's inequality, we can derive

$$
\begin{aligned}
\mathbb{E}_{q_0}\big(-\log p_\theta(x^{(0)})\big) =& \mathbb{E}_{q_0}\Big(-\log \int p_\theta(x^{(0:N)})\, dx^{(1:N)}\Big) \\
=& \mathbb{E}_{q_0}\Big(-\log \int \frac{p_\theta(x^{(0:N)})}{q(x^{(1:N)} \,|\, x^{(0)})} q(x^{(1:N)} \,|\, x^{(0)})\, dx^{(1:N)}\Big) \\
\leq& \mathbb{E}_{\mathbb{Q}^{(N)}}\Big(-\log \frac{p_\theta(x^{(0:N)})}{q(x^{(1:N)} \,|\, x^{(0)})}\Big) \\
=& \mathbb{E}_{\mathbb{Q}^{(N)}}\Big(-\log p(x^{(N)}) - \sum_{k=0}^{N-1} \log \frac{p_\theta(x^{(k)} \,|\, x^{(k+1)})}{q(x^{(k+1)} \,|\, x^{(k)})}\Big),
\end{aligned}
\tag{3}
$$

where $\mathbb{E}_{q_0}, \mathbb{E}_{\mathbb{Q}^{(N)}}$ denote the expectation with respect to the data distribution on $\mathcal{M}$, and the expectation with respect to the joint density $q(x^{(0:N)})$, respectively.

In order to derive an explicit training objective, we have to construct Markov chains on $\mathcal{M}$ with explicit transition densities. We discuss how this can be achieved in the next section.

We conclude this section by reformulating the variational bound (3) using relative entropy (see Song et al. (2021a) for a similar formulation of score-based diffusion models) . Recall that the relative entropy, or Kullback-Leibler (KL) divergence, from a probability density $Q_2$ to another probability density $Q_1$ on the same measure space, where $Q_1$ is absolutely continuous with respect to $Q_2$, is defined as $H(Q_1 \,|\, Q_2) := \mathbb{E}_{Q_1}\left( \log \frac{Q_1}{Q_2} \right)$. For simplicity, we also use the same notation for two probability measures. Adding the term $\mathbb{E}_{q_0}(\log q_0)$ to both sides of the inequality (3), we see that it is equivalent to (the data processing inequality)

$$H(q_0 \,|\, p_\theta) \leq H\big(\overleftarrow{\mathbb{Q}}^{(N)} \,|\, \mathbb{P}_\theta^{(N)}\big) \,, \tag{4}$$

where the upper bound is the relative entropy from the path measure $\mathbb{P}_\theta^{(N)}$ of the reverse process to the path measure $\overleftarrow{\mathbb{Q}}^{(N)}$ of the forward process (we include the arrow in the notation to indicate that paths of the forward process are viewed backwardly). Therefore, learning DDPMs using the variational bound (3) can be viewed as approximating probability measures in path space by the cross-entropy method (Zhang et al., 2014).

## 3 METHOD

### 3.1 PROJECTION SCHEME

We recall a projection scheme from Monte Carlo sampling methods on manifolds (Ciccotti et al., 2008; Zappa et al., 2018; Lelièvre et al., 2022), and we show that it allows us to construct Markov chains on $\mathcal{M}$ with tractable transition densities.

Given $x \in \mathcal{M}$ and a tangent vector $v \in T_x\mathcal{M}$ that is drawn from the standard Gaussian distribution on $T_x\mathcal{M}$, we compute the intermediate state $x' = x + \sigma^2 b(x) + \sigma v \in \mathbb{R}^n$, where $\sigma > 0$ is a positive constant and $b : \mathbb{R}^n \to \mathbb{R}^n$ is a smooth function. In general, $x'$ does not belong to $\mathcal{M}$. We consider the projection $y \in \mathcal{M}$ of $x'$ onto $\mathcal{M}$ along an orthogonal direction in the column space of $\nabla\xi(x)$. Precisely, the projected state $y$ is found by (numerically) solving the constraint equation for $c \in \mathbb{R}^{n-d}$

$$y = x + \sigma^2 b(x) + \sigma v + \nabla\xi(x)c, \;\; \text{such that} \;\; \xi(y) = 0 \in \mathbb{R}^{n-d} \,. \tag{5}$$

The choice of $b$ will affect the final invariant distribution and the convergence rate to equilibrium (see Section 3.5 for further discussion). There are $n - d$ constraints in equation 5 with the same number of unknown variables. In particular, when $\xi$ is scalar-valued, i.e. $n - d = 1$, solving equation 5 amounts to finding a root of a (nonlinear) scalar function.

In general, it is possible that for some vectors $v$ there are either no solution or multiple solutions to equation 5. When multiple solutions exist, we assume that the numerical solver finds one solution in a deterministic way. This is true as long as a deterministic solver is adopted with fixed initial condition $c = 0$. Let $\mathcal{F}_{x,\sigma}$ be the set of $v$ for which a solution can be found and denote by $\epsilon(x; \sigma) = \mathbb{P}(v \notin \mathcal{F}_{x,\sigma})$, i.e. the probability that no solution can be found. Since $\epsilon(x; \sigma) = 0$ when $\sigma = 0$ ($c = 0$ is a solution for any $v$), we can expect that $\epsilon(x; \sigma) \to 0$ as $\sigma$ decreases to zero. However, we do not require this assumption in deriving our method. We denote $\mathcal{M}_{x,\sigma}$ the set of all states in $\mathcal{M}$ that can be reached from $x$ by solving equation 5 with certain $v \in \mathcal{F}_{x,\sigma}$.

To derive the transition density of $y$ from $x$, we notice that, by applying the orthogonal projection matrix $P(x)$ to both sides of equation 5 and using the fact that $P(x)\nabla\xi(x) = 0$, we have the relation $\sigma v = P(x)(y - x - \sigma^2 b(x))$. This indicates that, given a state $x \in \mathcal{M}$ and $y \in \mathcal{M}_{x,\sigma}$, there is a unique tangent vector $v \in \mathcal{F}_{x,\sigma} \subseteq T_x\mathcal{M}$ that leads to $y$ by solving equation 5. In other words, the mapping from $v \in \mathcal{F}_{x,\sigma}$ to $y \in \mathcal{M}_{x,\sigma}$ is a bijection. Moreover, its inverse is explicitly given by

$$G_x : \mathcal{M}_{x,\sigma} \to \mathcal{F}_{x,\sigma} \subseteq T_x\mathcal{M}, \quad G_x(y) = \frac{1}{\sigma}P(x)(y - x - \sigma^2 b(x)) \,. \tag{6}$$

Recall that $U_x, U_y \in \mathbb{R}^{n \times d}$ denote the matrices whose columns form the orthonormal basis of $T_x\mathcal{M}$ and $T_y\mathcal{M}$, respectively. Using equation 6, we can derive (see Lelièvre et al. (2022) for more detailed discussions)

$$\det(DG_x(y)) = \sigma^{-d} \det(U_x^\top U_y) \,, \tag{7}$$

where the left hand side denotes the determinant of the Jacobian $DG_x(y) : T_y\mathcal{M} \to T_v T_x\mathcal{M} \cong \mathbb{R}^d$ of the map $G_x$ at $y$. Since $v$ is a Gaussian variable in $\mathcal{F}_{x,\sigma}$ (with a normalizing constant rescaled by $(1 - \epsilon(x; \sigma))^{-1}$), applying the change of variables formula for probability densities, we obtain the probability density of landing at $y$ from $x$:

$$
\begin{aligned}
q(y \mid x) &= (2\pi)^{-\frac{d}{2}} (1 - \epsilon(x; \sigma))^{-1} e^{-\frac{|P(x)(y - x - \sigma^2 b(x))|^2}{2\sigma^2}} | \det DG_x(y)| \\
&= (2\pi\sigma^2)^{-\frac{d}{2}} (1 - \epsilon(x; \sigma))^{-1} e^{-\frac{|P(x)(y - x - \sigma^2 b(x))|^2}{2\sigma^2}} | \det(U_x^\top U_y)|, \quad y \in \mathcal{M}_x.
\end{aligned}
\tag{8}
$$

## 3.2 Forward process

We construct the forward process in our model as a Markov chain on $\mathcal{M}$ whose transitions are defined by the projection scheme in equation 5. Specifically, given the current state $x^{(k)} \in \mathcal{M}$ at step $k$, where $k = 0, 1, \ldots, N-1$, the next state $x^{(k+1)} \in \mathcal{M}$ is determined by solving the constraint equation (for $c \in \mathbb{R}^{n-d}$):

$$
x^{(k+1)} = x^{(k)} + \sigma_k^2 b(x^{(k)}) + \sigma_k v^{(k)} + \nabla\xi(x^{(k)})c, \text{ such that } \xi(x^{(k+1)}) = 0 \in \mathbb{R}^{n-d}, \tag{9}
$$

where $\sigma_k > 0$ and $v^{(k)} \in T_{x^{(k)}}\mathcal{M}$ is a standard Gaussian variable in $T_{x^{(k)}}\mathcal{M}$. According to equations 1 and 8, we obtain the joint probability density of the forward process as

$$
q(x^{(1:N)} \mid x^{(0)}) = \prod_{k=0}^{N-1} q(x^{(k+1)} \mid x^{(k)}),
$$

$$
\text{where} \quad q(x^{(k+1)} \mid x^{(k)}) = (2\pi\sigma_k^2)^{-\frac{d}{2}} (1 - \epsilon(x^{(k)}; \sigma_k))^{-1} | \det(U_{x^{(k)}}^\top U_{x^{(k+1)}})| \tag{10}
$$

$$
\times \exp\left(-\frac{\left|P(x^{(k)})\left(x^{(k+1)} - x^{(k)} - \sigma_k^2 b(x^{(k)})\right)\right|^2}{2\sigma_k^2}\right).
$$

## 3.3 Reverse process

The reverse process in our model is a Markov chain on $\mathcal{M}$ whose transitions (from $x^{(k+1)}$ to $x^{(k)}$) are defined by the constraint equation

$$
x^{(k)} = x^{(k+1)} - \beta_{k+1}^2 b(x^{(k+1)}) + \beta_{k+1}^2 s^{(k+1),\theta}(x^{(k+1)}) + \beta_{k+1}\bar{v}^{(k+1)} + \nabla\xi(x^{(k+1)})c, \text{ such that } \xi(x^{(k)}) = 0, \tag{11}
$$

for $k = N-1, N-2, \ldots, 0$, where $\beta_{k+1} > 0$, $\bar{v}^{(k+1)}$ is a standard Gaussian variable in $T_{x^{(k+1)}}\mathcal{M}$, and $s^{(k+1),\theta}(x^{(k+1)}) \in \mathbb{R}^n$ depends on the learning parameter $\theta$. Combining equations 2 and 8, we obtain the joint probability density of the reverse process as

$$
p_\theta(x^{(0:N)}) = p(x^{(N)}) \prod_{k=0}^{N-1} p_\theta(x^{(k)} \mid x^{(k+1)}),
$$

$$
\text{where} \quad p_\theta(x^{(k)} \mid x^{(k+1)}) = (2\pi\beta_{k+1}^2)^{-\frac{d}{2}} \left(1 - \epsilon_\theta(x^{(k+1)}; \beta_{k+1})\right)^{-1} | \det(U_{x^{(k+1)}}^\top U_{x^{(k)}})|
$$

$$
\times \exp\left(-\frac{\left|P(x^{(k+1)})\left(x^{(k)} - x^{(k+1)} + \beta_{k+1}^2\left(b(x^{(k+1)}) - s^{(k+1),\theta}(x^{(k+1)})\right)\right)\right|^2}{2\beta_{k+1}^2}\right), \tag{12}
$$

and $\epsilon_\theta(x^{(k+1)}; \beta_{k+1})$ denotes the probability of having $\bar{v}^{(k+1)}$ with which no solution to (11) can be found.

## 3.4 Training objective

The training objective follows directly from the variational bound (3) on the negative log-likelihood, as well as the explicit expressions of transition densities in equations 10 and 12. Concretely, substituting equations 10 and 12 into the last line of (3), we get

$$
\mathbb{E}_{q_0}(-\log p_\theta(x^{(0)})) \leq \text{Loss}^{(N)}(\theta) + C^{(N)}, \tag{13}
$$

where

$$\text{Loss}^{(N)}(\theta) = \frac{1}{2}\mathbb{E}_{\mathbb{Q}^{(N)}}\left[\sum_{k=0}^{N-1}\beta_{k+1}^2\left|P(x^{(k+1)})\left(s^{(k+1),\theta}(x^{(k+1)}) - b(x^{(k+1)}) + \frac{x^{(k+1)} - x^{(k)}}{\beta_{k+1}^2}\right)\right|^2\right]$$

(14)

is our objective for training the parameter $\theta$ in the reverse process (recall that $\mathbb{E}_{\mathbb{Q}^{(N)}}$ denotes the expectation with respect to the forward process), the constant

$$C^{(N)} = -\mathbb{E}_{\mathbb{Q}^{(N)}}\left[\frac{1}{2}\sum_{k=0}^{N-1}\sigma_k^2\left|P(x^{(k)})\left(\frac{x^{(k+1)} - x^{(k)}}{\sigma_k^2} - b(x^{(k)})\right)\right|^2 + \log p(x^{(N)})\right] + d\sum_{k=0}^{N-1}\log\frac{\beta_{k+1}}{\sigma_k}$$

$$- \mathbb{E}_{\mathbb{Q}^{(N)}}\left(\sum_{k=0}^{N-1}\log\left(1 - \epsilon(x^{(k)};\sigma_k)\right)\right)$$

(15)

is independent of the training parameter $\theta$, and we have used $\log(1 - \epsilon_\theta(x^{(k+1)});\beta_{k+1}) \leq 0$ in deriving equation 13.

### 3.5 ALGORITHMIC DETAILS

The algorithms for training and data generation are summarized in Algorithms 1 and 2, respectively. Algorithm for sampling the forward process, which is similar to Algorithm 2, and algorithm for solving constraint equations are presented in Algorithms 3 and 4 in Appendix B. In the following, we discuss several algorithmic details of our method. Further details are left in Appendix B.

**Generation of trajectory data**  The optimal parameter $\theta$ is sought by minimizing the objective in equation 14, for which trajectory data of the forward process is required as training data. We sample trajectories of the forward process in a preparatory step before training, and train the model with min-batches sampled from this trajectory dataset. The trajectory dataset is updated by re-sampling trajectories every several training epochs (see line 2 and lines 10–12 in Algorithm 1).

**Choice of $N$, $\sigma_k$, and $\beta_{k+1}$**  The total number of steps $N$ should be large enough such that the forward Markov chain is able to reach equilibrium (approximately) starting from the data distribution. While larger $\sigma_k$, $\beta_{k+1}$ allow the Markov chains to make larger jumps, their size should be properly chosen (depending on the manifold) such that the solution to the constraint equations 9 and 11 can be found with high probability.

**Method for solving constraint equations 9 and 11**  As in Monte Carlo sampling methods on submanifolds (Ciccotti et al., 2008; Zappa et al., 2018; Lelièvre et al., 2022), we employ Newton's method to solve the constraint equations 9 and 11. This method has quadratic convergence (locally) and its implementation is simple. In most cases, a solution with high precision can be found within a few iteration steps (e.g. less than 5 steps). When no solution is found, one can re-generate the state or the entire trajectory. Our implementation of Newton's method is summarized in Algorithm 4.

**Choices of $b$ and sampling of the prior $p(x^{(N)})$**  For relatively simple manifolds, we can for simplicity set $b = 0$ and choose the prior (see line 2 in Algorithm 2) as the uniform distribution on $\mathcal{M}$. When $\mathcal{M}$ is non-compact or when the convergence of Markov chain to equilibrium is slow with $b = 0$, we can choose non-zero $b$ such as $b = -\nabla V$, i.e. the (full space) gradient of a function $V : \mathbb{R}^n \to \mathbb{R}$ in the ambient space. In this case, sampling the prior can be done by simulating a single long trajectory of the forward process. See Section 6.4 and Appendix B.4 for a concrete choice of nonzero $b$ in practice.

## 4 THEORETICAL RESULTS

In this section, we study the continuous-time limit of our proposed method. Let $T > 0$ and $g : [0, T] \to \mathbb{R}^+$ be a continuous function. Define $h = \frac{T}{N}$ and consider the case where $\sigma_k = \sqrt{h}g(kh)$,

---

**Algorithm 1** Training procedure

---

1: **Input**: training data $(y^i)_{1 \le i \le M}$, functions $(s^{(k+1),\theta}(x))_{0 \le k \le N-1}$, constants $\sigma_k, \beta_k > 0$, function $b : \mathbb{R}^n \to \mathbb{R}^n$, integer $N$, batch size $B > 0$, number of total training epochs $N_{\text{epoch}}$, integer $l_{\text{f}} > 0$, learning rate $r > 0$.
2: for each $x^{(0),i} = y^i$, generate a path $(x^{(0),i}, x^{(1),i}, \dots, x^{(N),i})$ using Algorithm 3
3: **for** $l = 1$ to $N_{\text{epoch}}$ **do**                                                      ▷ loop over epochs
4:     **for** $j = 1$ to $\lfloor M/B \rfloor$ **do**                                           ▷ loop over min-batches
5:         sample a min-batch $\mathcal{I} = (i_1, i_2, \dots, i_B)$ from the set of indices $\{1, 2, \dots, M\}$
6:         calculate loss:
7:         $\ell(\theta) = \frac{1}{2|\mathcal{I}|} \sum_{i \in \mathcal{I}} \sum_{k=0}^{N-1} \beta_{k+1}^2 \left| P(x^{(k+1),i}) \left( s^{(k+1),\theta}(x^{(k+1),i}) - b(x^{(k+1),i}) + \frac{x^{(k+1),i} - x^{(k),i}}{\beta_{k+1}^2} \right) \right|^2$
8:         $\theta = \text{optimizer\_update}(\theta, \ell(\theta), r)$                                  ▷ ADAM optimizer
9:     **end for**
10:     **if** $l \% l_{\text{f}} == 0$ **then**                                      ▷ update trajectories every $l_{\text{f}}$ epochs
11:         for each $x^{(0),i} = y^i$, re-generate paths $(x^{(0),i}, x^{(1),i}, \dots, x^{(N),i})$ using Algorithm 3
12:     **end if**
13: **end for**
14: **return** $\theta$

---

for $k = 0, 1, \dots, N-1$. It is shown in Ciccotti et al. (2008) that the forward process (9) converges strongly to the SDE on $\mathcal{M}$

$$dX_t = g^2(t)P(X_t)b(X_t)dt + g(t)dW_t^{\mathcal{M}}, \quad t \in [0, T], \tag{16}$$

where $W_t^{\mathcal{M}}$ is a Brownian motion over $\mathcal{M}$. Denote by $p(\cdot, t)$ the probability density of $X_t$ with respect to $\sigma_{\mathcal{M}}$ at time $t \in [0, T]$.

We have the following result, which characterizes the loss function in equation 14 as $N \to +\infty$.

**Theorem 4.1.** *Let $T > 0$ and $g : [0, T] \to \mathbb{R}^+$ be a continuous function. Define $h = \frac{T}{N}$ and $t_k = kh$, for $k = 0, 1, \dots, N-1$. Assume that $\sigma_k = \beta_{k+1} = \sqrt{h}g(t_k)$. Also assume that, for any parameter $\theta$, there is a $C^1$ function $s_\theta : \mathbb{R}^n \times [0, T] \to \mathbb{R}^n$ such that $s^{(k+1),\theta}(x) = s_\theta(x, t_{k+1})$ for all $k = 0, 1, \dots, N-1$ and $x \in \mathcal{M}$. For the loss function defined in equation 14, we have*

$$\lim_{N \to +\infty} \left( \text{Loss}^{(N)}(\theta) - \frac{Nd}{2} \right) = \mathbb{E}_{\mathbb{Q}} \left[ \frac{1}{2} \int_0^T \left| P(X_t)s_\theta(X_t, t) - \nabla_{\mathcal{M}} \log p(X_t, t) \right|^2 g^2(t) \, dt \right.$$

$$\left. + \int_0^T \left( P(X_t)b(X_t) - \frac{1}{2}\nabla_{\mathcal{M}} \log p(X_t, t) \right) \cdot \nabla_{\mathcal{M}} \log p(X_t, t) \, g^2(t) \, dt \right],$$

*where $\mathbb{E}_{\mathbb{Q}}$ on the right hand side denotes the expectation with respect to the paths of SDE (16) and $\nabla_{\mathcal{M}}$ denotes the gradient operator on $\mathcal{M}$.*

Based on Theorem 4.1, the variational bound (3), and its relative entropy formulation in (4), we obtain the following corollary, which elucidates the connection between our RDDPMs and Riemannian score-based generative models (De Bortoli et al., 2022) as $N \to +\infty$.

**Corollary 4.1.** *Under the same assumptions of Theorem 4.1, we have, for any parameter $\theta$,*

$$\lim_{N \to +\infty} H\left( \overleftarrow{\mathbb{Q}}^{(N)} \,|\, \mathbb{P}_\theta^{(N)} \right) = \frac{1}{2} \mathbb{E}_{\mathbb{Q}} \left[ \int_0^T \left| P(X_t)s_\theta(X_t, t) - \nabla_{\mathcal{M}} \log p(X_t, t) \right|^2 g^2(t) \, dt \right]$$

$$= H\left( \overleftarrow{\mathbb{Q}} \,|\, \mathbb{P}_\theta \right), \tag{17}$$

*where $\overleftarrow{\mathbb{Q}}$ denotes the path measure of the time-reversal of SDE (16), and $\mathbb{P}_\theta$ denotes the path measure of the SDE*

$$dY_t = g^2(T-t)P(Y_t)\left( -b(Y_t) + s_\theta(Y_t, T-t) \right)dt + g(T-t)dW_t^{\mathcal{M}}, \quad t \in [0, T], \tag{18}$$

*starting from $Y_0 = X_T$.*

The proofs of Theorem 4.1 and Corollary 4.1 are presented in Appendix A.

---

**Algorithm 2** Sampling trajectory of reverse process

---

1: **Input**: trained functions $(s^{(k+1),\theta}(x))_{0 \leq k \leq N-1}$, constants $\beta_k$, function $b : \mathbb{R}^n \to \mathbb{R}^n$, and integer $N$
2: draw sample $x^{(N)}$ from the prior distribution $p(x^{(N)})$
3: **for** $k = N - 1$ to $0$ **do**
4:     sample $\bar{z}^{(k+1)} \sim \mathcal{N}(0, I_n)$ and set $\bar{v}^{(k+1)} = P(x^{(k+1)})\bar{z}^{(k+1)}$
5:     set $x^{(k+\frac{1}{2})} = x^{(k+1)} + \beta_{k+1}^2 P(x^{(k+1)})\big(s^{(k+1),\theta}(x^{(k+1)}) - b(x^{(k+1)})\big) + \beta_{k+1}\bar{v}^{(k+1)}$
6:     $c, \text{flag} = \text{newton\_solver}(x^{(k+1)}, x^{(k+\frac{1}{2})}; \xi)$            $\triangleright$ solve (11) by Algorithm 4
7:     **if** flag == true **then**
8:         $x^{(k)} := x^{(k+\frac{1}{2})} + \nabla\xi(x^{(k+1)})c$
9:     **else**
10:         discard the trajectory and re-generate
11:     **end if**
12: **end for**
13: **return** $(x^{(N)}, x^{(N-1)}, \ldots, x^{(0)})$

---

## 5 RELATED WORK

**Denoising diffusion probabilistic models** DDPMs (Ho et al., 2020) employ a forward Markov chain to perturb data into noise and a reverse Markov chain to incrementally recover data from noise. The models are trained to minimize a variational bound on the negative log-likelihood. DDPMs have made remarkable achievement in generative modeling on Euclidean space. However, to our knowledge, prior to this study there was no successful extension of DDPMs to manifolds.

**Diffusion models on manifolds** Riemannian Score-based Generative Models (RSGMs;De Bortoli et al. (2022) extend SGMs to Riemannian manifolds. A major difficulty of RSGMs is due to the fact that the denoising score-matching objective involves heat kernel, which is not known analytically except for very special manifolds. In addition, RSGMs also require sampling of geodesic curves on manifolds. De Bortoli et al. (2022) and Lou et al. (2023) proposed to approximate heat kernel by eigenfunction expansion or Varadhan's approximation, but these approximations bring in extra errors. In contrast, our method requires neither geodesic curves nor heat kernel.

The Riemannian Diffusion Model (RDM;Huang et al. (2022)) adopts a variational diffusion model framework on Riemannian manifolds. Similar to our method, RDM framework considers submanifolds embedded in an Euclidean space and it also utilizes a variational upper bound on the negative log-likelihood as loss function. However, RDM requires to sample SDEs on manifolds using closest-point projection, which in general may be a difficult task. In contrast, the projection adopted in our method is applicable to general submanifolds and the computational cost is lower.

**Flow-based generative models on manifolds** Riemannian continuous normalizing flow models (Lou et al., 2020; Mathieu & Nickel, 2020; Rozen et al., 2021; Ben-Hamu et al., 2022; Chen & Lipman, 2024) extend the continuous-time flow framework (Chen et al., 2018) to manifolds. In particular, the methods proposed in Rozen et al. (2021); Ben-Hamu et al. (2022); Chen & Lipman (2024) are simulation-free for manifolds with simple geometry. The method in Chen & Lipman (2024) can deal with manifolds with general geometry, but it is not feasible for high-dimensional manifolds, where eigenfunctions of the Laplace-Beltrami operator are typically difficult to compute. Our method requires sampling trajectories in order to evaluate loss function. However, the cost for trajectory simulation can be alleviated by working with a pre-prepared trajectory set that is updated during training with a tunable frequency. Our method does not use eigenfunctions nor metrics and it can be easily applied to high-dimensional manifolds.

**Markov Chain Monte Carlo on submanifolds** Markov Chain Monte Carlo (MCMC) under equality constraints has been studied in several works (Zappa et al., 2018; Lelièvre et al., 2019; 2022). We adopt the same projection scheme from these works to develop our method.

## 6 EXPERIMENTS

We evaluate our method on datasets from earth and climate science, mesh data on learned manifolds, synthetic dataset of high-dimensional special orthogonal matrices, and dataset of molecular conformations under constraints. The last two novel datasets have not been studied by existing methods.

### 6.1 EARTH AND CLIMATE SCIENCE DATASETS

We utilize public datasets (NOAA, 2020a;b; Brakenridge, 2017; EOSDIS, 2020) on 2-D sphere, which are compiled by Mathieu & Nickel (2020). Table 1 summarizes the negative log-likelihood (NLL) of our method alongside results of prior methods. The learned densities are displayed in Figure 4 of Appendix B.1.

We point out a general issue regarding model evaluation on these datasets. We notice that in each dataset there are a few data points (i.e. isolated points) whose vicinity contains no other points, and the size of dataset is relatively small compared to its complex distribution. For such datasets, standard dataset splitting for cross-validation results in non-overlapping training/validation/test sets, whose empirical distributions are considerably different. As a consequence, computing the NLL on the test set either results in an overconfident assessment of the model (when the test set contains no isolated points) or requires evaluating the model on isolated points that are completely unseen during training (when the test set contains isolated points).

We argue that the phenomenon discussed above is general and should appear regardless of the methods employed. As a simple solution, we propose to identify the isolated points in validation and test sets and include them into the training set. In particular, evident improvement is achieved when isolated points are included in training set, as shown in the last line of Table 1. Further discussions are provided in Appendix B.1.

Table 1: Test negative log-likelihood (NLL) results on earth and climate science datasets. A smaller value indicates better performance.

|  | Volcano | Earthquake | Flood | Fire |
|---|---|---|---|---|
| Dataset size | 827 | 6120 | 4875 | 12809 |
| RCNF (Mathieu & Nickel, 2020) | $-6.05_{\pm 0.61}$ | $0.14_{\pm 0.23}$ | $1.11_{\pm 0.19}$ | $-0.80_{\pm 0.54}$ |
| Moser Flow (Rozen et al., 2021) | $-4.21_{\pm 0.17}$ | $-0.16_{\pm 0.06}$ | $0.57_{\pm 0.10}$ | $-1.28_{\pm 0.05}$ |
| CNFM (Ben-Hamu et al., 2022) | $-2.38_{\pm 0.17}$ | $-0.38_{\pm 0.01}$ | $0.25_{\pm 0.02}$ | $-1.40_{\pm 0.02}$ |
| RSGM (De Bortoli et al., 2022) | $-4.92_{\pm 0.25}$ | $-0.19_{\pm 0.07}$ | $0.48_{\pm 0.17}$ | $-1.33_{\pm 0.06}$ |
| RDM (Huang et al., 2022) | $-6.61_{\pm 0.96}$ | $-0.40_{\pm 0.05}$ | $0.43_{\pm 0.07}$ | $-1.38_{\pm 0.05}$ |
| RFM (Chen & Lipman, 2024) | $-7.93_{\pm 1.67}$ | $-0.28_{\pm 0.08}$ | $0.42_{\pm 0.05}$ | $-1.86_{\pm 0.11}$ |
| LogBM (Jo & Hwang, 2024) | $-9.52_{\pm 0.87}$ | $-0.30_{\pm 0.06}$ | $0.42_{\pm 0.08}$ | $-2.47_{\pm 0.11}$ |
| *Ours* | | | | |
| RDDPM | $-2.16_{\pm 1.92}$ | $-0.17_{\pm 0.10}$ | $0.49_{\pm 0.09}$ | $-1.48_{\pm 0.10}$ |
| RDDPM w/ isolated points | $-3.57_{\pm 1.05}$ | $-0.29_{\pm 0.04}$ | $0.43_{\pm 0.05}$ | $-1.56_{\pm 0.08}$ |

### 6.2 MESH DATA ON LEARNED MANIFOLDS

Our method can effectively handle manifolds with general geometries. For demonstration, we examine the Standard Bunny (Turk & Levoy, 1994) and Spot the Cow (Crane et al., 2013), two manifolds defined by triangle meshes. The datasets are created according to the $k$-th eigenfunction of the grid Laplacian operator using the same approach described in Jo & Hwang (2024); Chen & Lipman (2024).

Similar to Rozen et al. (2021), we first learn the function $\xi : \mathbb{R}^3 \to \mathbb{R}$, whose zero level set matches the manifold. We adopt the approach in Gropp et al. (2020), where $\xi$ is represented by a neural network and is trained such that on mesh data $\xi$ is close to zero and $|\nabla \xi|$ is close to one. Using this approach, we obtain a function $\xi$ whose value is at the order $10^{-2}$ on mesh data. Then, we perform a further refinement to the dataset such that all points belong to the learned manifold $\mathcal{M} = \{x \in$

$\mathbb{R}^3 | \xi(x) = 0\}$ up to a small error $10^{-5}$. The maximal distance between the original data and the refined data is smaller than $0.017$. Details for training $\xi$ and refining data are left in Appendix B.2.

We perform the training with the learned function $\xi$. Table 2 shows that our learned models achieve evident improvement over prior works in terms of NLL results. Figure 1 visualizes the agreement between generated samples (learned distributions) and datasets (target distributions).

Table 2: Test negative log-likelihood (NLL) results on mesh datasets. A smaller value indicates better performance.

| | Stanford Bunny | | Spot the Cow | |
|---|---|---|---|---|
| | $k = 50$ | $k = 100$ | $k = 50$ | $k = 100$ |
| RFM w/ Diff. (Chen & Lipman, 2024) | $1.48 \pm 0.01$ | $1.53 \pm 0.01$ | $0.95 \pm 0.05$ | $1.08 \pm 0.05$ |
| RFM w/ Bihar. (Chen & Lipman, 2024) | $1.55 \pm 0.01$ | $1.49 \pm 0.01$ | $1.08 \pm 0.05$ | $1.29 \pm 0.05$ |
| LogBM w/ Diff. (Jo & Hwang, 2024) | $1.42 \pm 0.01$ | $1.41 \pm 0.00$ | $0.99 \pm 0.03$ | $0.97 \pm 0.03$ |
| LogBM w/ Bihar. (Jo & Hwang, 2024) | $1.55 \pm 0.02$ | $1.45 \pm 0.01$ | $1.09 \pm 0.06$ | $0.97 \pm 0.02$ |
| *Ours* | | | | |
| RDDPM | $1.36 \pm 0.00$ | $1.31 \pm 0.01$ | $0.84 \pm 0.00$ | $0.77 \pm 0.00$ |

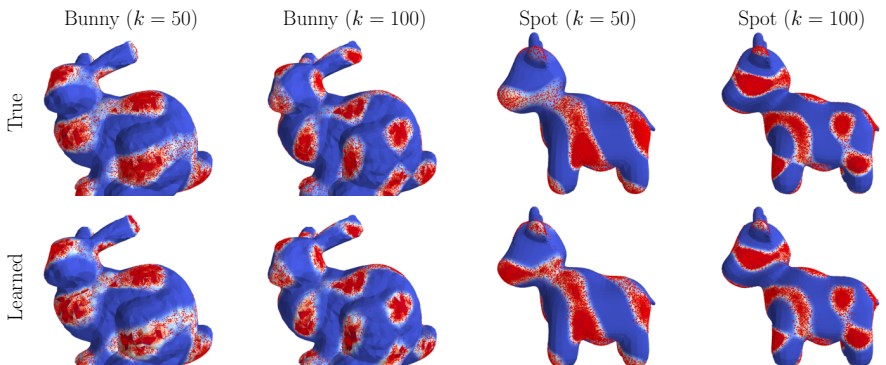

Figure 1: First row: datasets and true distributions. Second row: learned samples and distributions.

### 6.3 HIGH-DIMENSIONAL SPECIAL ORTHOGONAL GROUP

We apply our method to special orthogonal group $SO(10)$, viewed as a $45$-dimensional submanifold embedded in $\mathbb{R}^{100}$. The synthetic dataset is sampled from a multimodal distribution on $SO(10)$ with $m = 5$ modes. To assess the quality of generated data, we consider the statistics $\mathrm{tr}(S)$, $\mathrm{tr}(S^2)$, $\mathrm{tr}(S^4)$, and $\mathrm{tr}(S^5)$, where $\mathrm{tr}$ denotes the trace operator of matrices. Further details for the construction of the dataset and the choice of statistics are provided in Appendix B.3. Figure 2 indicates that our learned model can generate the data distribution accurately. What is more, the distributions of the forward process at intermediate steps are also faithfully reproduced.

### 6.4 ALANINE DIPEPTIDE

We apply our method to alanine dipeptide, a commonly studied model system in bio-physics. The configuration of the system can be characterized by its two dihedral angles $\phi$ and $\psi$ (see Figure 3a). In this study, we are interested in the configurations of the 10 non-hydrogen atoms of the system (in $\mathbb{R}^{30}$) with the fixed angle $\phi = -70°$.

Since the manifold is unbounded, we choose a nonzero function $b = -\nabla V$ in the forward process, where $V$ is proportional to the root mean squared deviation (RMSD) from a pre-selected reference configuration $x^{\mathrm{ref}}$. Accordingly, the prior distribution $p(x^{(N)})$ is a single-well distribution centered at $x^{\mathrm{ref}}$. Furthermore, we model $s^{(k+1),\theta}(x)$ in the reverse process using a network that preserves rotational equivariance and translational invariance. This, as well as our choice of $b$, guarantee that

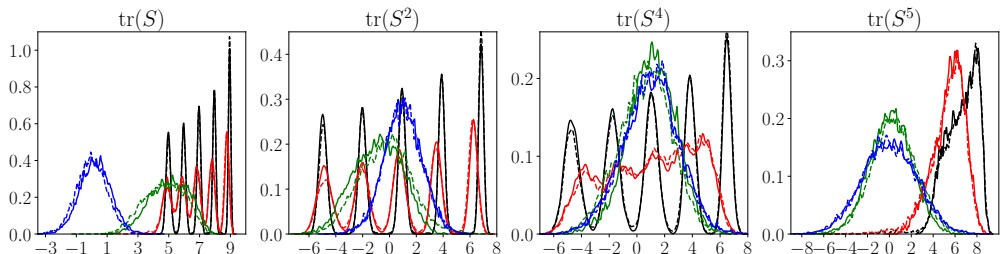

Figure 2: Results for $SO(10)$ with $m = 5$. Histograms of the statistics $\mathrm{tr}(S), \mathrm{tr}(S^2), \mathrm{tr}(S^4)$, and $\mathrm{tr}(S^5)$ for the forward process (solid line) and the learned reverse process (dashed line) at different steps $k = 0, 50, 200, 500$, colored in black, red, green, and blue, respectively.

the distribution $p_\theta(x^{(0)})$ generated by our model is invariant under $SE(3)$ (Xu et al., 2022). We refer to Appendix B.4 for implementation details and to Appendix C for theoretical support.

We employ three metrics to assess the quality of the generated configurations: the angle $\psi$, and two RMSDs (denoted by $\mathrm{RMSD}_1$ and $\mathrm{RMSD}_2$) with respect to two pre-defined reference configurations that are selected from two different wells. Figure 3b illustrates the histograms of these three metrics for the configurations generated by our model and the configurations in the dataset. The solid and dashed lines show the agreement between the distributions of the learned reverse process and the distributions of the forward process at different time steps $k$. In particular, the overlap between the lines in black, which correspond to step $k = 0$, demonstrates that the distribution of the generated samples (dashed) closely matches the data distribution (solid).

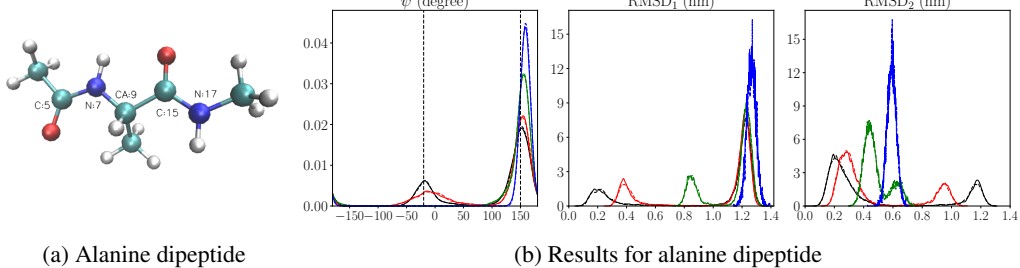

(a) Alanine dipeptide                    (b) Results for alanine dipeptide

Figure 3: (a) Illustration of the system. Names and 1-based indices are shown for atoms that define the dihedral angles. The dihedral angles $\phi$ and $\psi$ are defined by atoms whose 1-based indices are $5, 7, 9, 15$ and $7, 9, 15, 17$, respectively. (b) Histograms of the angle $\psi$, $\mathrm{RMSD}_1$, and $\mathrm{RMSD}_2$ for the forward process (solid line) and the learned reverse process (dashed line) at steps $k = 0, 10, 40, 200$ are colored in black, red, green, blue, respectively. The $\psi$ values of the two reference points that are used to define $\mathrm{RMSD}_1$ and $\mathrm{RMSD}_2$ are $-20°$ and $150°$, respectively (as shown by the two vertical dashed lines in the left panel).

# 7  CONCLUSION

We have proposed Riemannian Denoising Diffusion Probabilistic Models for generative modeling on submanifolds. Our method does not rely on sophisticated geometric objects on manifold and it is applicable to high-dimensional manifolds with nontrivial geometry. We have provided a theoretical analysis of our method in the continuous-time limit, which elucidates its connection to Riemannian score-based generative models. We have demonstrated the strong capability of our method on datasets from previous studies and from high-dimensional manifolds that can not be easily studied by existing methods.

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

## A  PROOFS OF THE CONTINUOUS-TIME LIMIT

In this section, we prove Theorem 4.1 and Corollary 4.1 in Section 4.

For notation simplicity, we denote by $\partial_i$ the derivative with respect to $x_i$ in the ambient space, and by $I$ the identity matrix of order $n$. We use subscripts to denote components of a vector and entries of a matrix. Also recall that the orthogonal projection matrix $P(x) \in \mathbb{R}^{n \times n}$ is well defined for $x \in \mathbb{R}^n$ and has the expression

$$P_{ij}(x) = \delta_{ij} - \sum_{\alpha,\alpha'=1}^{n-d} \partial_i \xi_\alpha(x) (\nabla \xi^\top \nabla \xi)^{-1}_{\alpha\alpha'}(x) \partial_j \xi_{\alpha'}(x), \quad 1 \le i, j \le n, \tag{19}$$

where $\delta_{ij}$ is the Dirac delta function.

First, we present the proof of Theorem 4.1.

*Proof of Theorem 4.1.* Let us write the forward process in equation 9 as

$$x^{(k+1)} = x^{(k+\frac{1}{2})} + \nabla \xi(x^{(k)}) c(x^{(k+\frac{1}{2})}),$$

where $x^{(k+\frac{1}{2})} = x^{(k)} + \sigma_k^2 b(x^{(k)}) + \sigma_k v^{(k)}$ and the dependence of $c$ on $x^{(k+\frac{1}{2})}$ is made explicit. Applying Lemma 1 at the end of this section, we obtain the expansion, for $1 \le i \le n$,

$$\begin{aligned}
&x_i^{(k+1)} \\
=&x_i^{(k)} + \sum_{j=1}^{n} P_{ij}(x^{(k)}) \Big( \sigma_k^2 b_j(x^{(k)}) + \sigma_k v_j^{(k)} \Big) \\
&+ \frac{1}{2} \sum_{j,l,r,r'=1}^{n} \Big( (I-P)_{ir} P_{r'l} \, \partial_{r'} P_{rj} \Big)(x^{(k)}) \Big( \sigma_k^2 b_j(x^{(k)}) + \sigma_k v_j^{(k)} \Big) \Big( \sigma_k^2 b_l(x^{(k)}) + \sigma_k v_l^{(k)} \Big) \\
&+ \frac{1}{6} \sum_{j,l,r=1}^{n} \sum_{\eta=1}^{n-d} \big( \partial_i \xi_\eta \, \partial_{jlr}^3 c_\eta \big)(x^{(k)}) \Big( x_j^{(k+\frac{1}{2})} - x_j^{(k)} \Big) \Big( x_l^{(k+\frac{1}{2})} - x_l^{(k)} \Big) \Big( x_r^{(k+\frac{1}{2})} - x_r^{(k)} \Big) \\
&+ o\big( |x^{(k+\frac{1}{2})} - x^{(k)}|^3 \big) \\
=&x_i^{(k)} + \sigma_k v_i^{(k)} + \sigma_k^2 \sum_{j=1}^{n} P_{ij}(x^{(k)}) b_j(x^{(k)}) + \frac{\sigma_k^2}{2} \sum_{j,r,r'=1}^{n} \big( (I-P)_{ir} \, \partial_{r'} P_{rj} \big)(x^{(k)}) v_j^{(k)} v_{r'}^{(k)} \\
&+ \sigma_k^3 R_i^{(k)} + o(\sigma_k^3), \tag{20}
\end{aligned}$$

where we have used the identity $\sum_{j=1}^{n} P_{ij}(x^{(k)}) v_j^{(k)} = v_i^{(k)}$ (since $v^{(k)}$ is a tangent vector), and $R_i^{(k)}$ is a term that satisfies $\sum_{i'=1}^{n} P_{ii'}(x^{(k)}) R_{i'}^{(k)} = 0$, for $1 \le i \le n$.

With the expansion above, we compute the loss function in equation 14. Using equation 20, the relation $\beta_{k+1} = \sigma_k = \sqrt{h} g(kh)$, and the assumption that $s^{(k+1),\theta}(x^{(k+1)}) = s_\theta(x^{(k+1)}, (k +$

$1)h) \in \mathbb{R}^n$, we can derive

$$\beta_{k+1}^2 \left| P(x^{(k+1)}) \left( s^{(k+1),\theta}(x^{(k+1)}) - b(x^{(k+1)}) + \frac{x^{(k+1)} - x^{(k)}}{\beta_{k+1}^2} \right) \right|^2$$

$$= \sigma_k^2 \sum_{i=1}^n \left| \sum_{i'=1}^n P_{ii'}(x^{(k+1)}) \left[ s_{\theta,i'}(x^{(k+1)}, (k+1)h) - b_{i'}(x^{(k+1)}) + \sum_{j=1}^n P_{i'j}(x^{(k)}) b_j(x^{(k)}) \right.\right.$$

$$\left.\left. + \frac{1}{2} \sum_{j,r,r'=1}^n \left((I-P)_{i'r}\, \partial_{r'} P_{rj}\right)(x^{(k)}) v_j^{(k)} v_{r'}^{(k)} + \frac{v_{i'}^{(k)}}{\sigma_k} + \sigma_k R_{i'}^{(k)} + o(\sigma_k) \right] \right|^2$$

$$= \mathcal{I}_1 + \mathcal{I}_2 + \mathcal{I}_3 + o(\sigma_k^2)\,, \tag{21}$$

where the three terms on the last line are defined as

$$\mathcal{I}_1 := \sigma_k^2 \sum_{i=1}^n \left| \sum_{i'=1}^n P_{ii'}(x^{(k+1)}) \left[ s_{\theta,i'}(x^{(k+1)}, (k+1)h) - b_{i'}(x^{(k+1)}) + \sum_{j=1}^n P_{i'j}(x^{(k)}) b_j(x^{(k)}) \right.\right.$$

$$\left.\left. + \frac{1}{2} \sum_{j,r,r'=1}^n \left((I-P)_{i'r}\, \partial_{r'} P_{rj}\right)(x^{(k)}) v_j^{(k)} v_{r'}^{(k)} + \sigma_k R_{i'}^{(k)} \right] \right|^2\,,$$

$$\mathcal{I}_2 := \sum_{i=1}^n \left( \sum_{i'=1}^n P_{ii'}(x^{(k+1)}) v_{i'}^{(k)} \right)^2\,,$$

$$\mathcal{I}_3 := 2\sigma_k \sum_{i,i'=1}^n P_{ii'}(x^{(k+1)}) \left[ s_{\theta,i'}(x^{(k+1)}, (k+1)h) - b_{i'}(x^{(k+1)}) + \sum_{j=1}^n P_{i'j}(x^{(k)}) b_j(x^{(k)}) \right.$$

$$\left. + \frac{1}{2} \sum_{j,r,r'=1}^n \left((I-P)_{i'r}\, \partial_{r'} P_{rj}\right)(x^{(k)}) v_j^{(k)} v_{r'}^{(k)} + \sigma_k R_{i'}^{(k)} \right] v_i^{(k)}\,,$$

respectively. In the following, we derive the expansions of the three terms above. For $\mathcal{I}_1$, expanding the functions $P, s_\theta, b$ using equation 20, we can derive

$$\mathcal{I}_1 = \sigma_k^2 \sum_{i=1}^n \left| \sum_{i'=1}^n P_{ii'}(x^{(k)}) \left[ s_{\theta,i'}(x^{(k)}, kh) - b_{i'}(x^{(k)}) + \sum_{j=1}^n P_{i'j}(x^{(k)}) b_j(x^{(k)}) \right.\right.$$

$$\left.\left. + \frac{1}{2} \sum_{j,r,r'=1}^n \left((I-P)_{i'r}\, \partial_{r'} P_{rj}\right)(x^{(k)}) v_j^{(k)} v_{r'}^{(k)} \right] + o(1) \right|^2$$

$$= \sigma_k^2 \sum_{i=1}^n \left| \sum_{i'=1}^n P_{ii'}(x^{(k)}) s_{\theta,i'}(x^{(k)}, kh) \right|^2 + o(\sigma_k^2)\,, \tag{22}$$

where we have used the relations $P^2 = P$ and $P(I-P) = 0$ satisfied by the orthogonal projection matrix $P$ to derive the second equality. For $\mathcal{I}_2$, using the relation $P^2 = P$ and equation 20, we can compute

$$\mathcal{I}_2 = \sum_{i,i'=1}^n P_{ii'}(x^{(k+1)}) v_i^{(k)} v_{i'}^{(k)}$$

$$= \sum_{i,i'=1}^n P_{ii'}(x^{(k)}) v_i^{(k)} v_{i'}^{(k)} + \sum_{i,i',r=1}^n \partial_r P_{ii'}(x^{(k)}) \left(x_r^{(k+1)} - x_r^{(k)}\right) v_i^{(k)} v_{i'}^{(k)}$$

$$+ \frac{1}{2} \sum_{i,i'=1}^n \sum_{r,r'=1}^n \partial_{rr'}^2 P_{ii'}(x^{(k)}) v_i^{(k)} v_{i'}^{(k)} (x_r^{(k+1)} - x_r^{(k)})(x_{r'}^{(k+1)} - x_{r'}^{(k)}) + o(|x^{(k+1)} - x^{(k)}|^2)$$

$$= |v^{(k)}|^2 + \sum_{i,i',r=1}^n \partial_r P_{ii'}(x^{(k)}) \left(x_r^{(k+1)} - x_r^{(k)}\right) v_i^{(k)} v_{i'}^{(k)}$$

$$+ \frac{\sigma_k^2}{2} \sum_{i,i'=1}^{n} \sum_{r,r'=1}^{n} \partial_{rr'}^2 P_{ii'}(x^{(k)}) v_i^{(k)} v_{i'}^{(k)} v_r^{(k)} v_{r'}^{(k)} + o(\sigma_k^2) \,. \tag{23}$$

Let's compute the three terms in equation 23. Using the expression of $P_{ii'}$ in (19), the fact that $\sum_{i=1}^{n} \partial_i \xi_\alpha(x^{(k)}) v_i^{(k)} = 0$, and the product rule, it is straightforward to verify that, for $1 \le r \le n$,

$$\sum_{i,i'=1}^{n} \partial_r P_{ii'}(x^{(k)}) v_i^{(k)} v_{i'}^{(k)} = - \sum_{i,i'=1}^{n} \partial_r \Big( \sum_{\alpha,\alpha'=1}^{n-d} \partial_i \xi_\alpha (\nabla\xi^\top \nabla\xi)_{\alpha\alpha'}^{-1} \partial_{i'} \xi_{\alpha'} \Big)(x^{(k)}) v_i^{(k)} v_{i'}^{(k)} = 0 \,. \tag{24}$$

Similarly, we can verify that

$$\sum_{i,i',r,r'=1}^{n} \partial_{rr'}^2 P_{ii'}(x^{(k)}) \, v_i^{(k)} v_{i'}^{(k)} v_r^{(k)} v_{r'}^{(k)}$$

$$= - 2 \sum_{i,i',r,r'=1}^{n} \sum_{\alpha,\alpha'=1}^{n-d} \Big( \partial_{ir}^2 \xi_\alpha (\nabla\xi^\top \nabla\xi)_{\alpha\alpha'}^{-1} \partial_{i'r'}^2 \xi_{\alpha'} \Big)(x^{(k)}) \, v_i^{(k)} v_{i'}^{(k)} v_r^{(k)} v_{r'}^{(k)} \,. \tag{25}$$

Hence, substituting equations 24–25 into equation 23, we obtain

$$\mathcal{I}_2 = |v^{(k)}|^2 - \sigma_k^2 \sum_{i,i',r,r'=1}^{n} \sum_{\alpha,\alpha'=1}^{n-d} \Big( \partial_{ir}^2 \xi_\alpha (\nabla\xi^\top \nabla\xi)_{\alpha\alpha'}^{-1} \partial_{i'r'}^2 \xi_{\alpha'} \Big)(x^{(k)}) \, v_i^{(k)} v_{i'}^{(k)} v_r^{(k)} v_{r'}^{(k)} + o(\sigma_k^2) \,. \tag{26}$$

For $\mathcal{I}_3$, we have

$$\mathcal{I}_3 = 2\sigma_k \sum_{i,i'=1}^{n} P_{ii'}(x^{(k)}) \bigg[ s_{\theta,i'}(x^{(k)}, kh) - b_{i'}(x^{(k)}) + \sum_{j=1}^{n} P_{i'j}(x^{(k)}) b_j(x^{(k)})$$

$$+ \frac{1}{2} \sum_{j,r,r'=1}^{n} \big( (I-P)_{i'r} \, \partial_{r'} P_{rj} \big)(x^{(k)}) v_j^{(k)} v_{r'}^{(k)} + \sigma_k R_{i'}^{(k)} \bigg] v_i^{(k)}$$

$$+ 2\sigma_k^2 \sum_{i,i',r=1}^{n} \partial_r (P_{ii'}(s_{\theta,i'} - b_{i'}))(x^{(k)}, kh) \, v_r^{(k)} v_i^{(k)}$$

$$+ 2\sigma_k^2 \sum_{i,i',r,j=1}^{n} \partial_r P_{ii'}(x^{(k)}) P_{i'j}(x^{(k)}) b_j(x^{(k)}) \, v_r^{(k)} v_i^{(k)}$$

$$+ \sigma_k^2 \sum_{i,i',r,j,r',j'=1}^{n} \partial_r P_{ii'}(x^{(k)}) \big( (I-P)_{i'j'} \, \partial_{r'} P_{j'j} \big)(x^{(k)}) v_j^{(k)} v_{r'}^{(k)} \, v_r^{(k)} v_i^{(k)} + o(\sigma_k^2)$$

$$= 2\sigma_k \sum_{i,i'=1}^{n} P_{ii'}(x^{(k)}) s_{\theta,i'}(x^{(k)}, kh) v_i^{(k)} + 2\sigma_k^2 \sum_{i,i',r=1}^{n} \partial_r (P_{ii'}(s_{\theta,i'} - b_{i'}))(x^{(k)}, kh) v_r^{(k)} v_i^{(k)}$$

$$+ 2\sigma_k^2 \sum_{i,i',r,j=1}^{n} \partial_r P_{ii'}(x^{(k)}) P_{i'j}(x^{(k)}) b_j(x^{(k)}) \, v_r^{(k)} v_i^{(k)}$$

$$+ \sigma_k^2 \sum_{i,i',r,j,r',j'=1}^{n} \partial_r P_{ii'}(x^{(k)}) \big( (I-P)_{i'j'} \, \partial_{r'} P_{j'j} \big)(x^{(k)}) v_j^{(k)} v_{r'}^{(k)} \, v_r^{(k)} v_i^{(k)} + o(\sigma_k^2) \,, \tag{27}$$

where we have used Taylor expansion with equation 20 and the fact that $|s_\theta(x, (k+1)h) - s_\theta(x, kh)| = O(h) = O(\sigma_k^2)$ to derive the first equality, and we have used the relations $P^2 = P$, $P(I-P) = 0$, and $\sum_{i'=1}^{n} P_{ii'}(x^{(k)}) R_{i'}^{(k)} = 0$ to derive the second equality. We further simplify the last two terms in the expression above. Notice that, similar to equation 24, we can verify that

$$\sum_{i,i',r=1}^{n} \partial_r P_{ii'}(x^{(k)}) P_{i'j}(x^{(k)}) v_r^{(k)} v_i^{(k)} = 0 \,, \quad 1 \le j \le n \,. \tag{28}$$

For the last term in equation 27, we can derive

$$\sum_{i,i',r,j,r',j'=1}^{n} \partial_r P_{ii'}(x^{(k)})\big((I-P)_{i'j'}\,\partial_{r'}P_{j'j}\big)(x^{(k)})v_j^{(k)}v_{r'}^{(k)}\,v_r^{(k)}v_i^{(k)}$$

$$= \sum_{i',r,j,r',j'=1}^{n} \big(\partial_r P_{i'j'}\,\partial_{r'}P_{j'j}\big)(x^{(k)})v_j^{(k)}v_{r'}^{(k)}\,v_r^{(k)}v_{i'}^{(k)}$$

$$= \sum_{i',r,j,r',j'=1}^{n} \Bigg[ \sum_{\alpha,\alpha',\eta,\eta'=1}^{n-d} \big(\partial_{i'r}^2\xi_\alpha(\nabla\xi^\top\nabla\xi)_{\alpha\alpha'}^{-1}\partial_{j'}\xi_{\alpha'}\big)\big(\partial_{jr'}^2\xi_\eta(\nabla\xi^\top\nabla\xi)_{\eta\eta'}^{-1}\partial_{j'}\xi_{\eta'}\big)\Bigg](x^{(k)})v_j^{(k)}v_{r'}^{(k)}\,v_r^{(k)}v_{i'}^{(k)}$$

$$= \sum_{i',r,j,r'=1}^{n} \Bigg[ \sum_{\alpha,\eta=1}^{n-d} \big(\partial_{i'r}^2\xi_\alpha(\nabla\xi^\top\nabla\xi)_{\alpha\eta}^{-1}\partial_{jr'}^2\xi_\eta\big)\Bigg](x^{(k)})v_j^{(k)}v_{r'}^{(k)}\,v_r^{(k)}v_{i'}^{(k)}\,,$$

(29)

where the first equation follows by applying the product rule to the identity $P(I-P)=0$ and using the relation $\sum_{i=1}^{n} P_{ii'}v_i^{(k)} = v_{i'}^{(k)}$, the second equation follows from the expression (19) and the fact that several terms vanish due to the orthogonality relation $\sum_{i=1}^{n} \partial_i\xi_\alpha(x^{(k)})v_i^{(k)} = 0$, and the last equation follows from the fact that $\sum_{j'=1}^{n} \partial_{j'}\xi_{\alpha'}\partial_{j'}\xi_{\eta'} = (\nabla\xi^\top\nabla\xi)_{\alpha'\eta'}$. Combining equations 27, 28, and 29, we obtain

$$\mathcal{I}_3 = 2\sigma_k \sum_{i,i'=1}^{n} P_{ii'}(x^{(k)})s_{\theta,i'}(x^{(k)},kh)v_i^{(k)} + 2\sigma_k^2 \sum_{i,i',r=1}^{n} \partial_r(P_{ii'}(s_{\theta,i'}-b_{i'}))(x^{(k)},kh)v_r^{(k)}v_i^{(k)}$$

$$+ \sigma_k^2 \sum_{i',r,j,r'=1}^{n} \Bigg[ \sum_{\alpha,\eta=1}^{n-d} \big(\partial_{i'r}^2\xi_\alpha(\nabla\xi^\top\nabla\xi)_{\alpha\eta}^{-1}\partial_{jr'}^2\xi_\eta\big)\Bigg](x^{(k)})v_j^{(k)}v_{r'}^{(k)}\,v_r^{(k)}v_{i'}^{(k)} + o(\sigma_k^2)\,. \quad (30)$$

Substituting equations 22, 26, and 30 into equation 21, we obtain (after cancellation of terms in $\mathcal{I}_2$ and $\mathcal{I}_3$)

$$\beta_{k+1}^2\Bigg|P(x^{(k+1)})\Big(s^{(k+1),\theta}(x^{(k+1)}) - b(x^{(k+1)}) + \frac{x^{(k+1)}-x^{(k)}}{\beta_{k+1}^2}\Big)\Bigg|^2$$

$$= \mathcal{I}_1 + \mathcal{I}_2 + \mathcal{I}_3 + o(\sigma_k^2)$$

$$= \sigma_k^2\big|P(x^{(k)})s_\theta(x^{(k)},kh)\big|^2 + \big|v^{(k)}\big|^2 + 2\sigma_k^2 \sum_{i,i',r=1}^{n} \partial_r(P_{ii'}(s_{\theta,i'}-b_{i'}))(x^{(k)},kh)v_r^{(k)}v_i^{(k)} \quad (31)$$

$$+ 2\sigma_k \sum_{i,i'=1}^{n} P_{ii'}(x^{(k)})s_{\theta,i'}(x^{(k)},kh)v_i^{(k)} + o(\sigma_k^2)\,.$$

Since $v^{(k)}$ is a standard Gaussian random variable in $T_{x^{(k)}}\mathcal{M}$, taking expectation in equation 31 and substituting it into equation 14, we obtain

$$\mathrm{Loss}^{(N)}(\theta) = \frac{Nd}{2} + \mathbb{E}_{\mathbb{Q}^{(N)}}\Bigg[ \sum_{k=0}^{N-1} \sigma_k^2\Big(\frac{1}{2}\big|P(x^{(k)})s_\theta(x^{(k)},kh)\big|^2$$

$$+ \sum_{i,i',r=1}^{n} \partial_r(P_{ii'}(s_{\theta,i'}-b_{i'}))(x^{(k)},kh)\,P_{ri}(x^{(k)})\Big)\Bigg] + o(1)\,.$$

(32)

In the above, we have used the identities $\mathbb{E}(v_i^{(k)}) = 0$, $\mathbb{E}(|v^{(k)}|^2) = d$, and $\mathbb{E}(v_r^{(k)}v_i^{(k)}) = P_{ri}(x^{(k)})$, where the last one can be verified using the fact that $v^{(k)} = P(x^{(k)})z^{(k)}$, with $z^{(k)}$ being a standard Gaussian random variable in $\mathbb{R}^n$.

Taking the limit $N \to +\infty$, and using the fact that the forward process $x^{(k)}$ converges to the SDE (16) (Ciccotti et al., 2008), we can derive

$$\lim_{N\to+\infty} \Big(\mathrm{Loss}^{(N)}(\theta) - \frac{Nd}{2}\Big)$$

$$= \lim_{N \to +\infty} \mathbb{E}_{\mathbb{Q}^{(N)}} \left[ \sum_{k=0}^{N-1} \sigma_k^2 \left( \frac{1}{2} |P(x^{(k)}) s_\theta(x^{(k)}, kh)|^2 + \sum_{i,i',r=1}^{n} \partial_r (P_{ii'}(s_{\theta,i'} - b_{i'}))(x^{(k)}, kh) P_{ri}(x^{(k)})) \right) \right]$$

$$= \mathbb{E}_{\mathbb{Q}} \int_0^T \left[ \frac{1}{2} |P(X_t) s_\theta(X_t, t)|^2 + \mathrm{div}_{\mathcal{M}} \big( P(s_\theta - b) \big)(X_t, t) \right] g^2(t) dt$$

$$= \int_0^T \left[ \int_{\mathcal{M}} \left( \frac{1}{2} |P(x) s_\theta(x, t)|^2 + \mathrm{div}_{\mathcal{M}} \big( P(s_\theta - b) \big)(x, t) \right) p(x, t) \, d\sigma_{\mathcal{M}}(x) \right] g^2(t) \, dt$$

$$= \int_0^T \left[ \int_{\mathcal{M}} \left( \frac{1}{2} |P(x) s_\theta(x, t)|^2 - P(x) \big( s_\theta(x, t) - b(x) \big) \cdot \nabla_{\mathcal{M}} \log p(x, t) \right) p(x, t) \, d\sigma_{\mathcal{M}}(x) \right] g^2(t) \, dt$$

$$= \frac{1}{2} \int_0^T \left[ \int_{\mathcal{M}} |P(x) s_\theta(x, t) - \nabla_{\mathcal{M}} \log p(x, t)|^2 p(x, t) \, d\sigma_{\mathcal{M}}(x) \right] g^2(t) \, dt$$

$$+ \int_0^T \left[ \int_{\mathcal{M}} \left( (P(x) b(x) - \frac{1}{2} \nabla_{\mathcal{M}} \log p(x, t)) \cdot \nabla_{\mathcal{M}} \log p(x, t) \right) p(x, t) \, d\sigma_{\mathcal{M}}(x) \right] g^2(t) \, dt$$

$$= \mathbb{E}_{\mathbb{Q}} \left[ \frac{1}{2} \int_0^T |P(X_t) s_\theta(X_t, t) - \nabla_{\mathcal{M}} \log p(X_t, t)|^2 g^2(t) \, dt \right.$$

$$\left. + \int_0^T \left( P(X_t) b(X_t) - \frac{1}{2} \nabla_{\mathcal{M}} \log p(X_t, t) \right) \cdot \nabla_{\mathcal{M}} \log p(X_t, t) \, g^2(t) \, dt \right],$$

where we have used integration by parts on $\mathcal{M}$, and the expression $\mathrm{div}_{\mathcal{M}} f = \sum_{i,r=1}^{n} P_{ir} \partial_r f_i$ for $f : \mathcal{M} \to \mathbb{R}^n$ (which can be verified using Lemma A.1 in Zhang (2020)). $\qquad\square$

Next, we present the proof of Corollary 4.1.

*Proof of Corollary 4.1.* Using the assumption $\beta_{k+1} = \sigma_k$, the projection scheme in equation 9 and the relation $P(x^{(k)}) \nabla \xi(x^{(k)}) = 0$, we can simplify the constant $C^{(N)}$ in equation 15 as

$$C^{(N)} = -\mathbb{E}_{\mathbb{Q}^{(N)}} \left( \log p(x^{(N)}) + \frac{1}{2} \sum_{k=0}^{N-1} |v^{(k)}|^2 + \sum_{k=0}^{N-1} \log \big( 1 - \epsilon(x^{(k)}; \sigma_k) \big) \right). \tag{33}$$

Therefore, using the definition of relative entropy (see equation 4), the loss function in equation 14, the constant $C^{(N)}$ in equation 15, and applying Theorem 4.1, we have

$$\lim_{N \to +\infty} H\big( \overleftarrow{\mathbb{Q}}^{(N)} \,|\, \mathbb{P}_\theta^{(N)} \big)$$

$$= \lim_{N \to +\infty} \left[ \mathrm{Loss}^{(N)}(\theta) + C^{(N)} + \mathbb{E}_{\mathbb{Q}^{(N)}} \left( \sum_{k=0}^{N-1} \log \big( 1 - \epsilon_\theta(x^{(k+1)}; \sigma_k) \big) \right) \right] + \mathbb{E}_{q_0} (\log q_0)$$

$$= \lim_{N \to +\infty} \left[ \mathrm{Loss}^{(N)}(\theta) - \mathbb{E}_{\mathbb{Q}^{(N)}} \left( \log p(x^{(N)}) + \frac{1}{2} \sum_{k=0}^{N-1} |v^{(k)}|^2 + \sum_{k=0}^{N-1} \log \frac{1 - \epsilon(x^{(k)}; \sigma_k)}{1 - \epsilon_\theta(x^{(k+1)}; \sigma_k)} \right) \right] + \mathbb{E}_{q_0} (\log q_0)$$

$$= \lim_{N \to +\infty} \left( \mathrm{Loss}^{(N)}(\theta) - \mathbb{E}_{\mathbb{Q}^{(N)}} \log p(x^{(N)}) - \frac{Nd}{2} \right) + \mathbb{E}_{q_0} (\log q_0)$$

$$= \mathbb{E}_{\mathbb{Q}} \left[ \log p(X_0, 0) - \log p(X_T, T) + \frac{1}{2} \int_0^T |P(X_t) s_\theta(X_t, t) - \nabla_{\mathcal{M}} \log p(X_t, t)|^2 g^2(t) \, dt \right.$$

$$\left. + \int_0^T \left( P(X_t) b(X_t) - \frac{1}{2} \nabla_{\mathcal{M}} \log p(X_t, t) \right) \cdot \nabla_{\mathcal{M}} \log p(X_t, t) \, g^2(t) \, dt \right],$$

$$\tag{34}$$

where the third equality follows because the terms containing $\epsilon(x^{(k)}; \sigma_k)$ and $\epsilon_\theta(x^{(k+1)}; \sigma_k)$ vanish and $\lim_{N \to +\infty} \mathbb{E}_{\mathbb{Q}^{(N)}} (\sum_{k=0}^{N-1} |v^{(k)}|^2 - Nd) = 0$, both of which can be verified using the asymptotic expression of complementary error function. Note that the density $p(x, t)$ of SDE (16) solves the Fokker-Planck equation

$$\frac{\partial p}{\partial t}(x, t) = -g^2(t) \mathrm{div}_{\mathcal{M}} \big( P(x) b(x) p(x, t) \big) + \frac{g^2(t)}{2} \Delta_{\mathcal{M}} p(x, t), \quad x \in \mathcal{M}, \quad t \in [0, T]. \tag{35}$$

Therefore, we have

$$
\mathbb{E}_{\mathbb{Q}}\Big[ \log p(X_0,0) - \log p(X_T,T) \Big]
$$

$$
= \int_{\mathcal{M}} \log p(x,0) p(x,0)\, d\sigma_{\mathcal{M}}(x) - \int_{\mathcal{M}} \log p(x,T) p(x,T)\, d\sigma_{\mathcal{M}}(x)
$$

$$
= -\int_0^T \frac{d}{dt}\Big( \int_{\mathcal{M}} \log p(x,t)\, p(x,t)\, d\sigma_{\mathcal{M}}(x)\Big)\, dt
$$

$$
= -\int_0^T \Big[ \int_{\mathcal{M}} \big( \log p(x,t) + 1\big)\frac{\partial p}{\partial t}(x,t)\, d\sigma_{\mathcal{M}}(x)\Big]\, dt
$$

$$
= -\int_0^T \Big[ \int_{\mathcal{M}} \big( \log p(x,t) + 1\big)\Big( - g^2(t)\mathrm{div}_{\mathcal{M}}\big(P(x)b(x)p(x,t)\big) + \frac{g^2(t)}{2}\Delta_{\mathcal{M}} p(x,t)\Big) d\sigma_{\mathcal{M}}(x)\Big]\, dt
$$

$$
= -\int_0^T \Big[ \int_{\mathcal{M}} \Big( \big(P(x)b(x) - \frac{1}{2}\nabla_{\mathcal{M}} \log p(x,t)\big)\cdot \nabla_{\mathcal{M}} \log p(x,t)\Big)\, p(x,t)\, d\sigma_{\mathcal{M}}(x)\Big] g^2(t)\, dt
$$

$$
= -\mathbb{E}_{\mathbb{Q}}\Big[ \int_0^T \big( P(X_t)b(X_t) - \frac{1}{2}\nabla_{\mathcal{M}} \log p(X_t,t)\big)\cdot \nabla_{\mathcal{M}} \log p(X_t,t)\, g^2(t)\, dt\Big],
$$

$$
\tag{36}
$$

where we have used equation 35 to derive the fourth equality and integration by parts on $\mathcal{M}$ to derive the fifth equality. Combining equations 34 and 36, we obtain

$$
\lim_{N\to+\infty} H\big(\overleftarrow{\mathbb{Q}}^{(N)} \,|\, \mathbb{P}_\theta^{(N)}\big) = \mathbb{E}_{\mathbb{Q}}\Big[ \frac{1}{2}\int_0^T \big| P(X_t)s_\theta(X_t,t) - \nabla_{\mathcal{M}} \log p(X_t,t)\big|^2 g^2(t)\, dt\Big]. \tag{37}
$$

Finally, note that $\overleftarrow{\mathbb{Q}}$ is the path measure of the time-reversal $Y_t = X_{T-t}$ of SDE (16), which satisfies (De Bortoli et al., 2022, Theorem 3.1)

$$
dY_t = g^2(T-t)\Big( -P(Y_t)b(X_t) + \nabla_{\mathcal{M}} \log p(Y_t, T-t)\Big) dt + g(T-t)dW_t^{\mathcal{M}}, \quad t\in[0,T], \tag{38}
$$

and $\mathbb{P}_\theta$ is the path measure of SDE (18). Applying Girsanov's theorem (Hsu, 2002, Theorem 8.1.2), we obtain

$$
\begin{aligned}
\frac{d\mathbb{P}_\theta}{d\overleftarrow{\mathbb{Q}}} = \exp\Big( &\int_0^T g^2(T-t)\big(P(Y_t)s_\theta(Y_t,T-t) - \nabla_{\mathcal{M}} \log p(Y_t,T-t)\big)\cdot dW_t^{\mathcal{M}}\\
&- \frac{1}{2}\int_0^T \big| P(Y_t)s_\theta(Y_t,T-t) - \nabla_{\mathcal{M}} \log p(Y_t,T-t)\big|^2 g^2(T-t)\, dt\Big),
\end{aligned} \tag{39}
$$

where $W_t^{\mathcal{M}}$ is a Brownian motion on $\mathcal{M}$ under $\overleftarrow{\mathbb{Q}}$. Therefore, we have

$$
\begin{aligned}
H(\overleftarrow{\mathbb{Q}} \,|\, \mathbb{P}_\theta) =& \mathbb{E}_{\overleftarrow{\mathbb{Q}}}\Big( \log \frac{d\overleftarrow{\mathbb{Q}}}{d\mathbb{P}_\theta}\Big)\\
=& \mathbb{E}_{\overleftarrow{\mathbb{Q}}}\Big( \frac{1}{2}\int_0^T \big| P(Y_t)s_\theta(Y_t,T-t) - \nabla_{\mathcal{M}} \log p(Y_t,T-t)\big|^2 g^2(T-t)\, dt\Big)\\
=& \mathbb{E}_{\mathbb{Q}}\Big( \frac{1}{2}\int_0^T \big| P(X_t)s_\theta(X_t,t) - \nabla_{\mathcal{M}} \log p(X_t,t)\big|^2 g^2(t)\, dt\Big),
\end{aligned} \tag{40}
$$

where the second equality follows from the fact that the stochastic integration in equation 39 vanishes after taking logarithm and expectation, and the third equality follows by a change of variable $t \leftarrow T-t$ and the fact that $Y_t = X_{T-t}$. The conclusion is obtained after combining equations 37 and 40. $\qquad\square$

Finally, we present the technical lemma on the projection scheme in equation 5, which was used in the proof of Theorem 4.1.

**Lemma 1.** *Given $x \in \mathcal{M}$ and $x' \in \mathbb{R}^n$, the solution to the problem*

$$
y = x' + \nabla\xi(x)c(x'), \quad c(x') \in \mathbb{R}^{n-d}, \ \ such\ that\ \ \xi(y) = 0 \tag{41}
$$

*has the following expansion as $x'$ approaches to $x$*

$$\partial_j c_\eta(x) = -\sum_{\alpha=1}^{n-d}(\nabla\xi^\top\nabla\xi)_{\eta\alpha}^{-1}(x)\partial_j\xi_\alpha(x)\,, \quad 1 \le j \le n\,,$$

$$\partial_{jl}^2 c_\eta(x) = \sum_{\alpha=1}^{n-d}(\nabla\xi^\top\nabla\xi)_{\eta\alpha}^{-1}(x)\sum_{r,r'=1}^{n}\Big(\partial_r\xi_\alpha\partial_{r'}P_{rj}\,P_{r'l}\Big)(x)\,, \quad 1 \le j,l \le n\,,$$

(42)

*for $1 \le \eta \le n-d$. Moreover, as $x'$ approaches to $x$, the following expansion of $y$ in equation 41 holds*

$$y_i = x_i + \sum_{j=1}^{n}P_{ij}(x)(x_j'-x_j) + \frac{1}{2}\sum_{j,l=1}^{n}\Big[\sum_{r,r'=1}^{n}((I-P)_{ir}P_{r'l}\partial_{r'}P_{rj})(x)\Big](x_j'-x_j)(x_l'-x_l)$$

$$+ \frac{1}{6}\sum_{j,l,r=1}^{n}\Big(\sum_{\eta=1}^{n-d}\partial_i\xi_\eta(x)\,\partial_{jlr}^3 c_\eta(x)\Big)(x_j'-x_j)(x_l'-x_l)(x_r'-x_r) + o(|x'-x|^3)\,,$$

(43)

*where $1 \le i \le n$.*

*Proof.* Differentiating (with respect to $x'$) the constraint equation

$$\xi_\alpha(x' + \nabla\xi(x)c(x')) = 0\,, \quad \alpha = 1,\ldots,n-d\,,$$

we get

$$\sum_{r=1}^{n}\partial_r\xi_\alpha\big(x' + \nabla\xi(x)c(x')\big)\Big(\delta_{rj} + \sum_{\eta=1}^{n-d}\partial_r\xi_\eta(x)\partial_j c_\eta(x')\Big) = 0\,, \quad 1 \le j \le n\,. \quad (44)$$

Setting $x' = x$ in equation 44 (notice that $c(x') = 0$ when $x' = x$) and multiplying both sides by $(\nabla\xi^\top\nabla\xi)^{-1}(x)$, we obtain the first identity in equation 42. In particular, using equation 19, we have

$$\delta_{rj} + \sum_{\eta=1}^{n-d}\partial_r\xi_\eta(x)\partial_j c_\eta(x) = \delta_{rj} - \sum_{\eta,\alpha=1}^{n-d}\Big(\partial_r\xi_\eta(\nabla\xi^\top\nabla\xi)_{\eta\alpha}^{-1}\partial_j\xi_\alpha\Big)(x) = P_{rj}(x)\,, \quad 1 \le r,j \le n\,.$$

(45)

Next, we show the second identity in equation 42. Differentiating equation 44 again, setting $x' = x$ and using equation 45, we get, for $1 \le \alpha \le n-d$ and $1 \le j,l \le n$,

$$0 = \sum_{r,r'=1}^{n}\partial_{rr'}^2\xi_\alpha(x)\Big(\delta_{rj} + \sum_{\eta=1}^{n-d}\partial_r\xi_\eta(x)\partial_j c_\eta(x)\Big)\Big(\delta_{r'l} + \sum_{\eta=1}^{n-d}\partial_{r'}\xi_\eta(x)\partial_l c_\eta(x)\Big)$$

$$+ \sum_{r=1}^{n}\partial_r\xi_\alpha(x)\Big(\sum_{\eta=1}^{n-d}\partial_r\xi_\eta(x)\partial_{jl}^2 c_\eta(x)\Big)$$

$$= \sum_{r,r'=1}^{n}\Big(\partial_{rr'}^2\xi_\alpha P_{rj}P_{r'l}\Big)(x) + \sum_{\eta=1}^{n-d}\Big((\nabla\xi^\top\nabla\xi)_{\alpha\eta}\partial_{jl}^2 c_\eta\Big)(x)\,,$$

from which we can solve, for $1 \le \eta \le n-d$ and $1 \le j,l \le n$,

$$\partial_{jl}^2 c_\eta(x) = -\sum_{\alpha=1}^{n-d}\sum_{r,r'=1}^{n}\Big((\nabla\xi^\top\nabla\xi)_{\eta\alpha}^{-1}\partial_{rr'}^2\xi_\alpha\,P_{rj}P_{r'l}\Big)(x)$$

$$= \sum_{\alpha=1}^{n-d}\sum_{r,r'=1}^{n}\Big((\nabla\xi^\top\nabla\xi)_{\eta\alpha}^{-1}\partial_r\xi_\alpha\partial_{r'}P_{rj}\,P_{r'l}\Big)(x)\,,$$

where the second equality follows from the product rule and the identity $\sum_{r=1}^{n}P_{rj}\partial_r\xi_\alpha = 0$. This shows the second identity in equation 42.

Lastly, we prove the expansion in equation 43. Note that the second identity in equation 42 and equation 19 implies

$$\sum_{\eta=1}^{n-d}(\partial_i\xi_\eta\partial_{jl}^2 c_\eta)(x) = \sum_{r,r'=1}^{n}\big((I-P)_{ir}P_{r'l}\,\partial_{r'}P_{rj}\big)(x), \quad 1\le i,j,l\le n\,. \tag{46}$$

By expanding $c(x')$ at $x'=x$ to the third order, noticing that $c(x)=0$, and using equations 45 and 46 for the first and second order derivatives respectively, we can derive

$$y_i = x'_i + \sum_{\eta=1}^{n-d}\partial_i\xi_\eta(x)c_\eta(x')$$

$$= x_i + (x'_i - x_i) + \sum_{\eta=1}^{n-d}\partial_i\xi_\eta(x)\bigg[\sum_{j=1}^{n}\partial_j c_\eta(x)(x'_j-x_j) + \frac{1}{2}\sum_{j,l=1}^{n}\partial_{jl}^2 c_\eta(x)(x'_j-x_j)(x'_l-x_l)$$

$$+ \frac{1}{6}\sum_{j,l,r=1}^{n}\partial_{jlr}^3 c_\eta(x)\,(x'_j-x_j)(x'_l-x_l)(x'_r-x_r)\bigg] + o(|x'-x|^3)$$

$$= x_i + \sum_{j=1}^{n}P_{ij}(x)(x'_j-x_j) + \frac{1}{2}\sum_{j,l=1}^{n}\bigg[\sum_{r,r'=1}^{n}\big((I-P)_{ir}\partial_{r'}P_{rj}P_{r'l}\big)(x)\bigg](x'_j-x_j)(x'_l-x_l)$$

$$+ \frac{1}{6}\sum_{j,l,r=1}^{n}\bigg(\sum_{\eta=1}^{n-d}\partial_i\xi_\eta(x)\partial_{jlr}^3 c_\eta(x)\bigg)(x'_j-x_j)(x'_l-x_l)(x'_r-x_r) + o(|x'-x|^3)\,,$$

which proves equation 43. $\qquad\square$

## B  Details of Algorithms and Experiments

We present the algorithms for sampling the forward process and for solving constraint equations in Algorithms 3 and 4, respectively. In the following, we discuss several further algorithmic details (see Section 3.5) that are common in our experiments. Specific details of each experiment are discussed in the subsections below.

**Neural networks and training setup**  As described in Theorem 4.1, the functions $(s^{(k+1),\theta}(x))_{0\le k\le N-1}$ are represented by a single function $s_\theta(x,t)$ with parameter $\theta$, which is in turn modeled by a multilayer perceptron (MLP). We employ SiLU as the activation function. We do not require that the output of the neural network belongs to the tangent space, thanks to the presence of the projection in both the forward and the reverse processes. Alternative strategies for designing neural networks with outputs in tangent space are proposed in De Bortoli et al. (2022).

We train our models using PyTorch, where we employ the Adam optimizer with fixed learning rate $r = 5\times 10^{-4}$ and we clip the gradients of the parameters when the 2-norm exceeds 10.0. We also implement an exponential moving average for the model weights (Polyak & Juditsky, 1992) with a decay rate of 0.999. All experiments are run on a single NVIDIA A40 GPU with 48G memory.

**NLL calculation**  Following De Bortoli et al. (2022) and Chen & Lipman (2024), for each experiment we train our model in five different runs with random seed values ranging from 0 to 4. In each run, the dataset is divided into training, validation, and test sets with ratio 80:10:10. We compute the test NLL (i.e. NLL on test set) using the model that yields the best validation NLL during the training. On both validation set and test set, the NLL is calculated by (see the second line in (3))

$$-\log p_\theta(x^{(0)}) = -\log\mathbb{E}_{\mathbb{Q}^{(N)}}\bigg[p(x^{(N)})\prod_{k=0}^{N-1}\frac{p_\theta(x^{(k)}\,|\,x^{(k+1)})}{q(x^{(k+1)}\,|\,x^{(k)})}\bigg|x^{(0)}\bigg], \tag{47}$$

where the conditional expectation is estimated by sampling multiple (50 or 100) paths of the forward process starting from each $x^{(0)}$, and we replace both $\epsilon(x^{(k)};\sigma_k)$ in equation 10 and $\epsilon_\theta(x^{(k+1)};\beta_{k+1})$ in equation 12 by zero (which is valid since $\sigma_k$ and $\beta_{k+1}$ are small in our experiment). We use equation 47 to compute NLLs in the experiments where the prior distribution $p(x^{(N)})$ is uniform distribution.

---

**Algorithm 3** Sampling trajectory of forward process

---

1: **Input**: $x^{(0)} \in \mathcal{M}$, constants $\sigma_k$, function $b : \mathbb{R}^n \to \mathbb{R}^n$, and integer $N$
2: **for** $k = 0$ to $N - 1$ **do**
3:      generate $z^{(k)} \sim \mathcal{N}(0, I_n)$ and set $v^k = P(x^{(k)})z^{(k)}$
4:      set $x^{(k+\frac{1}{2})} := x^{(k)} + \sigma_k^2 b(x^{(k)}) + \sigma_k v^{(k)}$
5:      $c, \text{flag} = \text{newton\_solver}(x^{(k)}, x^{(k+\frac{1}{2})}; \xi)$.             $\triangleright$ solve (9) by Algorithm 4
6:      **if** flag == true **then**
7:          set $x^{(k+1)} := x^{(k+\frac{1}{2})} + \nabla\xi(x^{(k)})c$
8:      **else**
9:          discard the trajectory and re-generate
10:      **end if**
11: **end for**
12: **return** $(x^{(0)}, x^{(1)}, \ldots, x^{(N)})$

---

**Algorithm 4** newton\_solver$(x, x'; \xi)$             $\triangleright$ solve $\xi(x' + \nabla\xi(x)c) = 0$ by Newton's method

---

1: **Input**: $x \in \mathcal{M}$, $x' \in \mathbb{R}^n$, $\xi : \mathbb{R}^n \to \mathbb{R}^{n-d}$, maximal iteration steps $n_{\text{step}}$, tolerance $\text{tol} > 0$
2: **Initialization**: set $c = 0 \in \mathbb{R}^{n-d}$ and flag=false
3: **for** $k = 0$ to $n_{\text{step}} - 1$ **do**
4:      Solve linear equation $\left[ \nabla\xi\left(x' + \nabla\xi(x)c\right)^\top \nabla\xi(x) \right] u = -\xi\left(x' + \nabla\xi(x)c\right)$ for $u \in \mathbb{R}^{n-d}$
5:      $c \leftarrow c + u$
6:      **if** $|\xi(x' + \nabla\xi(x)c)| < \text{tol}$ **then**
7:          set flag=true, and go to Step 10
8:      **end if**
9: **end for**
10: **return** c, flag

---

**Model parameters** As in Theorem 4.1, we choose $T > 0$, integer $N > 0$, function $g(t) = \gamma_{\min} + \frac{t}{T}(\gamma_{\max} - \gamma_{\min})$ for some $\gamma_{\max} \geq \gamma_{\min} > 0$, and parameters $\sigma_k = \beta_{k+1} = \sqrt{h}g(kh)$, where $h = \frac{T}{N}$ and $k = 0, 1, \ldots, N - 1$. We choose $b = 0$ in all experiments except the alanine dipeptide.

**Generation of standard Gaussian variables in** $T_x\mathcal{M}$. Let $z$ be a standard Gaussian random variable in $\mathbb{R}^n$. It is straightforward to verify that $v = P(x)z$ is a standard Gaussian random variable in $T_x\mathcal{M}$, where $P(x)$ is the orthogonal projection matrix to the tangent space $T_x\mathcal{M}$. We use this fact to generate tangent vectors in the forward and reverse processes (see line 3 of Algorithm 3 and line 4 of Algorithm 2, respectively).

Values of all the parameters in our experiments are summarized in Table 3.

## B.1 EARTH AND CLIMATE SCIENCE DATASETS

The unit sphere is viewed as a submanifold of $\mathbb{R}^3$, which is defined by the zero level set of the function $\xi(x) = |x| - 1$, for $x \in \mathbb{R}^3$. Newton's method is not necessary for the projection steps (9) and (11), since the general scheme (5) has the closed-form solution:

$$y = \left(1 - |\sigma^2 P(x)b(x) + \sigma v|^2\right)^{\frac{1}{2}} x + \sigma^2 P(x)b(x) + \sigma v, \tag{48}$$

as long as $|\sigma^2 P(x)b(x) + \sigma v| < 1$.

In the following, we present a careful study on the dataset splitting and the NLL calculation for these datasets. In fact, the datasets in this example contain isolated data points and the distributions of the data points are complex. Consequently, the training, validation, and test sets obtained from the standard data splitting method exhibit significant differences in their distributions, leading to inaccurate evaluation results and an increased risk of overfitting. Specifically, the presence of isolated data points in the validation set or in the test set (therefore not in the training set) can significantly affect the final test results. Figure 5a illustrates this phenomenon in the case of the volcano dataset,

Table 3: Parameters in our experiments. $\gamma_{\min}, \gamma_{\max}, N, T$ are the parameters in our model; $l_{\mathrm{f}}, N_{\mathrm{epoch}}, B$ are the parameters in Algorithm 1; $N_{\mathrm{node}}, N_{\mathrm{layer}}$ are the numbers of the hidden nodes per layer and the hidden layers of the neural networks, respectively.

| Datasets | $\gamma_{\min}$ | $\gamma_{\max}$ | $N$ | $T$ | $l_{\mathrm{f}}$ | $N_{\mathrm{epoch}}$ | $B$ | $N_{\mathrm{node}}$ | $N_{\mathrm{layer}}$ |
|---|---|---|---|---|---|---|---|---|---|
| Volcano | 0.01 | 1.0 | 400 | 4.0 | 1 | 20000 | 128 | 512 | 5 |
| Earthquake | 0.01 | 1.0 | 400 | 4.0 | 1 | 20000 | 512 | 512 | 5 |
| Flood | 0.01 | 1.0 | 400 | 4.0 | 1 | 20000 | 512 | 512 | 5 |
| Fire | 0.01 | 1.0 | 400 | 4.0 | 1 | 20000 | 512 | 512 | 5 |
| Bunny, $k = 50$ | 0.07 | 0.07 | 800 | 8.0 | 100 | 2000 | 2048 | 256 | 5 |
| Bunny, $k = 100$ | 0.07 | 0.07 | 500 | 5.0 | 100 | 2000 | 2048 | 256 | 5 |
| Spot, $k = 50$ | 0.1 | 0.1 | 500 | 5.0 | 100 | 2000 | 2048 | 256 | 5 |
| Spot, $k = 100$ | 0.1 | 0.1 | 300 | 3.0 | 100 | 2000 | 2048 | 256 | 5 |
| SO(10), $m = 3$ | 0.2 | 2.0 | 500 | 1.0 | 100 | 2000 | 512 | 512 | 3 |
| SO(10), $m = 5$ | 0.2 | 2.0 | 500 | 1.0 | 100 | 2000 | 512 | 512 | 3 |
| Alanine dipeptide | 1.0 | 1.0 | 200 | 0.1 | 100 | 5000 | 512 | 512 | 5 |

where the model is not trained on points in the vicinity of the three (one) isolated points in the test (validation) set, since there is no training data in that area. Accordingly, the loss and NLL values computed on different sets are significantly different, as shown by the solid lines in Figures 5b–5c.

To resolve this issue, we propose to include the isolated points in the validation set and the test set to the training set. These points can be identified by binning the data points according to their latitude and longitude values. Figure 5b shows that the NLL values computed on different sets become closer within a single run when the isolated points are included in the training set, and Figure 5c shows that the NLL values become smaller (better) among five different runs.

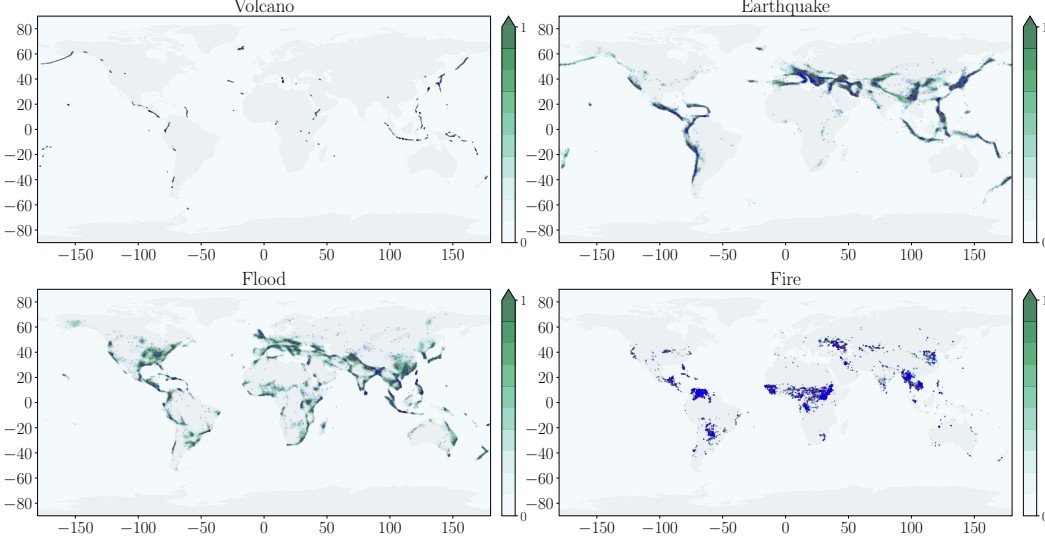

Figure 4: The learned densities on earth and climate science datasets, with the standard dataset splitting. Darker green color indicates areas of higher likelihood. Red dots and blue dots show points in test set and generated samples, respectively.

### B.2 MESH DATA ON LEARNED MANIFOLDS

To create the datasets, we adopt the approach described by Jo & Hwang (2024) and Chen & Lipman (2024). The data is generated according to the density defined by the $k$-th clamped eigenfunction of the Laplacian operator on a mesh that has been upsampled threefold.

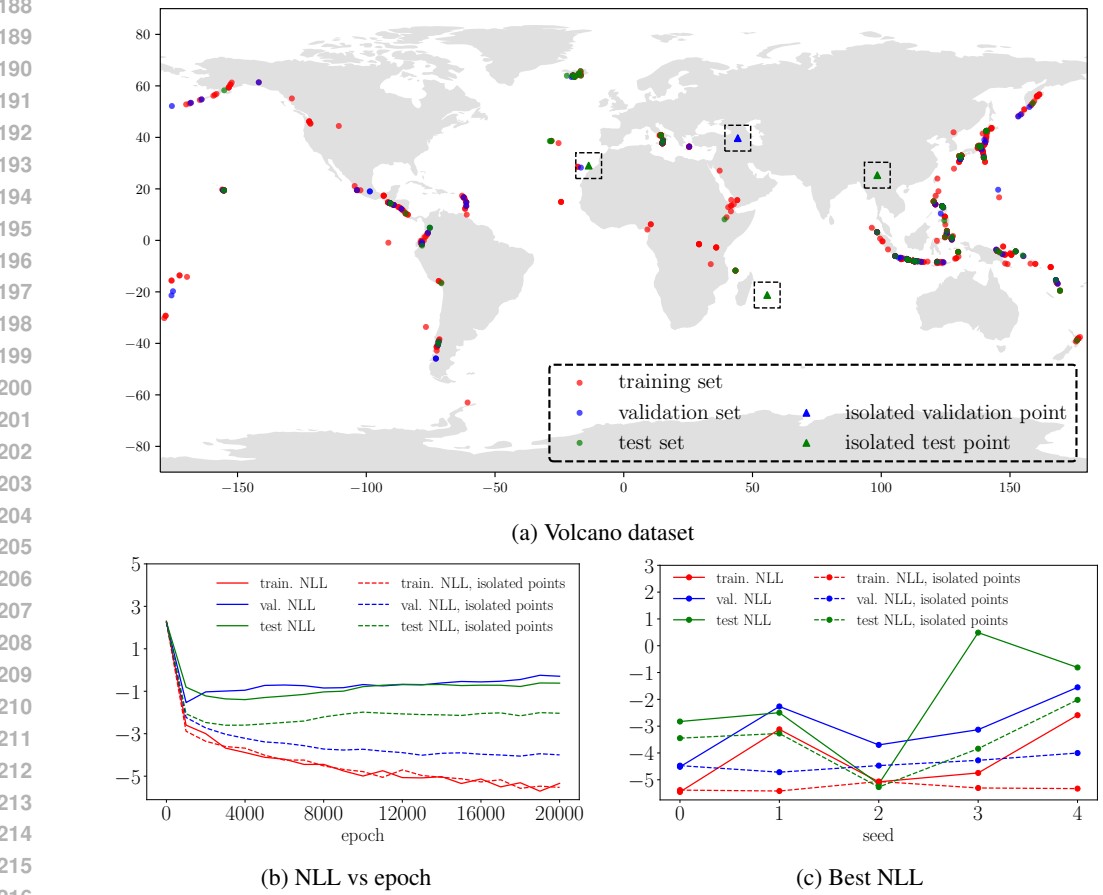

(a) Volcano dataset

(b) NLL vs epoch

(c) Best NLL

Figure 5: Volcano dataset. (a) Red, blue, and green points represent the training set, validation set, and test set, respectively, obtained from the standard dataset splitting with random seed 4. Triangles indicate the isolated points in the validation set and in the test set. (b) The training NLL, the validation NLL, and the test NLL during the training with random seed 4. The solid lines correspond to the training where the standard dataset splitting is employed. The dashed lines correspond to the training where the isolated points (triangles in (a)) are included in the training set. (c) The best NLLs for five different runs with random seeds 0, 1, 2, 3, and 4.

The function $\xi : \mathbb{R}^3 \to \mathbb{R}$ is modeled by a MLP with 3 hidden layers, each of which has 128 nodes. Different from the activation function in our model, here we use Softplus activation function, where the parameter $\beta$ is set to 10. The loss function for learning $\xi$ is

$$\ell(\xi) = \frac{1}{|\mathcal{D}|} \sum_{x \in \mathcal{D}} |\xi(x)| + \frac{\lambda}{|\mathcal{D}'|} \sum_{y \in \mathcal{D}'} (|\nabla \xi(y)| - 1)^2, \tag{49}$$

where $\lambda = 0.1$, $\mathcal{D}$ denotes the set of vertices of a high-resolution mesh, and the set $\mathcal{D}'$ contains samples near the manifolds that are obtained by perturbing samples $x \in \mathcal{D}$ according to $y = x + c\epsilon$, with $\epsilon \sim \mathcal{N}(0, I_3)$ and $c = 0.05$. The first term in equation 49 imposes that $\xi$ is close to zero on vertices, whereas the second term serves as a regularization term and ensures that $\xi$ has non-vanishing gradient near the manifold. The neural network is trained for 200000 steps using Adam optimizer, with batch size 512 and learning rate $10^{-4}$.

With the learned function $\xi$, we consider the manifold defined by $\mathcal{M} = \{x \in \mathbb{R}^3 | \xi(x) = 0\}$. The values of $\xi$ on the dataset are at the order $10^{-2}$. To ensure that the data is on $\mathcal{M}$ with high precision, we refine the dataset by solving the following ordinary differential equation (ODE):

$$\frac{dx_t}{dt} = -\xi(x_t) \nabla \xi(x_t), \quad t \geq 0, \tag{50}$$

starting from each point in the dataset until the condition $|\xi(x_t)| < 10^{-5}$ is reached (notice that equation 50 is a gradient flow and $\lim_{t \to \infty} |\xi(x_t)| = 0$). This ensures that the refined points conform to the manifold accurately.

For the Newton's method in generating paths, we set $\text{tol} = 10^{-4}$ and $n_{\text{step}} = 10$ in Algorithm 4. Due to the complex geometry of the objects, there is a small portion of paths (less than $1.0\%$) that can not be successfully generated. Apart from these paths, the Newton's method reaches convergence within 3 iteration steps.

### B.3 HIGH-DIMENSIONAL SPECIAL ORTHOGONAL GROUP

We view the group $\text{SO}(10)$ as a 45-dimensional submanifold of $\mathbb{R}^{100}$ that corresponds to (a connected component of) the zero level set of the map $\xi : \mathbb{R}^{100} \to \mathbb{R}^{55}$, whose components consist of the upper triangle portion of the matrix $S^\top S - I_{10}$, where $S$ is a $10 \times 10$ matrix.

The dataset is constructed as a mixture of $m$ wrapped normal distributions, each of which is the image (under the exponential map) of a normal distribution in the tangent space of a center $S_i \in \text{SO}(10)$, $1 \leq i \leq m$. To ensure multimodality, we define the centers $S_i$ as follows. We initially define a $2 \times 2$ matrix $A_0 := \begin{bmatrix} \cos \frac{\pi}{3} & \sin \frac{\pi}{3} \\ -\sin \frac{\pi}{3} & \cos \frac{\pi}{3} \end{bmatrix}$, which represents a rotation by $\frac{\pi}{3}$ radians. We then construct block diagonal matrices of order 10 by incorporating $A_0$ and the identity matrix $I_2$ in various combinations:

$$X_1 = \text{diag}\{A_0, I_2, I_2, I_2, I_2\}, \quad X_2 = \text{diag}\{A_0, A_0, I_2, I_2, I_2\}, \quad X_3 = \text{diag}\{A_0, A_0, A_0, I_2, I_2\},$$
$$X_4 = \text{diag}\{A_0, A_0, A_0, A_0, I_2\}, \quad X_5 = \text{diag}\{A_0, A_0, A_0, A_0, A_0\}. \tag{51}$$

The centers $S_i$ of the $m$ wrapped normal distributions are chosen as $S_i = Q_i^\top X_i Q_i$, where $Q_i \in \text{SO}(10)$ are randomly drawn from the uniform distribution. According to equation 51, the statistics $\eta(S) = (\text{tr}(S), \text{tr}(S^2), \text{tr}(S^4), \text{tr}(S^5))$ of the centers can be explicitly computed (using the trace identities $\text{tr}(AB) = \text{tr}(BA)$ and $\text{tr}(Q_i^\top X_i Q_i) = \text{tr}(X_i)$) as

$$\eta(S_1) = (9, 7, 7, 9), \quad \eta(S_2) = (8, 4, 4, 8), \quad \eta(S_3) = (7, 1, 1, 7),$$
$$\eta(S_4) = (6, -2, -2, 6), \quad \eta(S_5) = (5, -5, -5, 5). \tag{52}$$

To generate data in the dataset, we select a center $S_i$ with equal probability, sample tangent vectors $Y$ from the normal distribution (in the tangent space at $S_i$) with zero mean and standard deviation 0.05, and then compute their images $S$ under the exponential map, that is, $S = S_i e^{S_i^\top Y}$.

Due to our choice of $A_0$, the distribution of $\text{tr}(S^3)$ is narrowly concentrated at its local peaks. To see this, using $S_i^3 = Q_i^\top X_i^3 Q_i$ and the fact that $S_i^\top Y$ is anti-symmetric, we can compute

$$\begin{aligned} \text{tr}(S^3) &= \text{tr}(S_i e^{S_i^\top Y} S_i e^{S_i^\top Y} S_i e^{S_i^\top Y}) \\ &= \text{tr}\left(S_i^3 + 3(S_i^3)S_i^\top Y + O(|Y|^2)\right) \\ &= \text{tr}(S_i^3) + 3\,\text{tr}\left(X_i^3 Q_i S_i^\top Y Q_i^\top\right) + O(|Y|^2) \\ &= \text{tr}(S_i^3) + O(|Y|^2), \end{aligned} \tag{53}$$

where we have used the trace identity $\text{tr}(AB) = \text{tr}(BA)$, the expansion of matrix exponential, and the fact that the diagonal elements of $X_i^3 Q_i S_i^\top Y Q_i^\top$ are all zero (hence its trace equals zero), since $X_i^3$ is a diagonal matrix and $Q_i S_i^\top Y Q_i^\top$ is anti-symmetric. Therefore, we omit $\text{tr}(S^3)$ in our choice of statistical analysis for clear presentation.

For the Newton's method, we set $\text{tol} = 10^{-6}$ and $n_{\text{step}} = 10$ in Algorithm 4. For this example, the convergence is always reached within 3 iteration steps.

We also examine the dataset with $m = 3$ modes, in which case we use centers defined as $S_i = Q_i^\top X_i Q_i$ for $i = 1, 2, 3$ (the random matrices $Q_i$ are different from those in the case of $m = 5$). Figure 6 shows that the distributions of the learned reverse processes match the corresponding distributions of the forward process at different Markov chain jump steps.

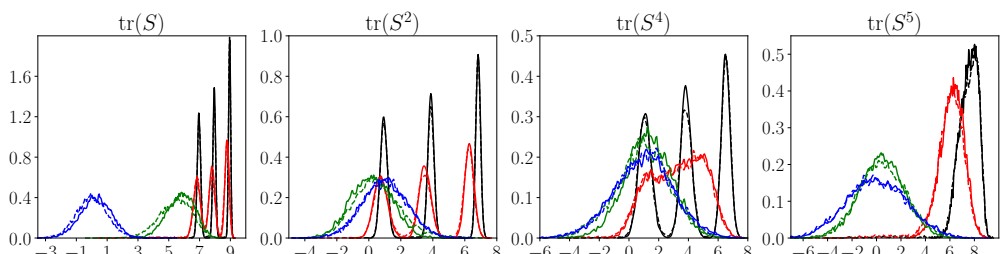

Figure 6: Results for SO(10) with $m = 3$. Histograms of the statistics $\mathrm{tr}(S)$, $\mathrm{tr}(S^2)$, $\mathrm{tr}(S^4)$, and $\mathrm{tr}(S^5)$ for the forward process (solid line) and the learned reverse process (dashed line) at different steps. Colors black, red, green, and blue correspond to steps $k = 0, 50, 200, 500$, respectively.

## B.4 ALANINE DIPEPTIDE

To generate the dataset, we initially perform a constrained molecular simulation of alanine dipeptide in water for 1ns using the molecular dynamics package GROMACS (Van Der Spoel et al., 2005) with step-size 1fs. We apply the harmonic biasing method in COLVARS module (Fiorin et al., 2013), where the collective variable is chosen as the dihedral angle $\phi$ and the harmonic potential is centered at $\phi = -70°$ with the force constant 5.0. Further simulation details are omitted since they are similar to those in Lelièvre et al. (2024). In total, $10^4$ configurations are obtained by recording every 100 simulation steps. We exclude the hydrogen atoms and work with the coordinates of the 10 non-hydrogen atoms in the system (see Figure 3a). In a final preparatory step, we apply the refinement technique in Appendix B.2 (see equation 50) to the recorded coordinates, so that the data in the dataset lives in the manifold $\mathcal{M} = \{x \in \mathbb{R}^{30} | \phi(x) = -70°\}$ up to a small numerical error of order $10^{-5}$.

Since $\mathcal{M}$ is unbounded, we adopt a nonzero function $b$ in our model to make sure that the Markov chain processes stay in bounded region. To this end, we choose a reference configuration $x^{\mathrm{ref}}$ from the dataset and define the potential function

$$V(x) = \frac{\kappa}{2}|R_x^*(x - w_x^*) - x^{\mathrm{ref}}|^2, \quad x \in \mathbb{R}^{30}, \tag{54}$$

with $\kappa = 50$, where $R_x^*$, $w_x^*$ are the optimal rotation and the optimal translation that minimize the RMSD (see equation 56). The function $b$ is defined as (the negative gradient of $V$ in full space)

$$b = -\nabla V(x) = -\kappa \left(R_x^*(x - w_x^*) - x^{\mathrm{ref}}\right), \tag{55}$$

where the second equality follows by differentiating $V$ in equation 54 and using the first order optimality equations satisfied by $R_x^*$ and $w_x^*$ (also see Coutsias et al. (2004)).

We also build our model to make sure that the generated distribution is SE(3)-invariant (i.e. invariant under rotations and translations). For this, we rely on the theoretical results in Xu et al. (2022) and in Appendix C.

One can check that $V(x)$ is SE(3)-invariant and $b$ satisfies property (#) in Appendix C, that is, $b$ is equivariant under rotations and invariant under translations. This guarantees that the prior distribution $p(x^{(N)})$, which we choose as the invariant distribution of the forward process, is SE(3)-invariant as well.

We still need to make sure that the transition densities of the reverse Markov chain are SE(3)-invariant. For this purpose, in the reverse process we set $s^{(k+1),\theta}(x) = (R_x^*)^\top f_\theta(R_x^*(x - w_x^*), \frac{(k+1)T}{N})$, where $f_\theta : \mathbb{R}^{30} \times \mathbb{R} \to \mathbb{R}^{30}$ is modeled by a single MLP with parameter $\theta$, and both $R_x^*$ and $b_x^*$ are computed by the Kabsch algorithm (Kabsch, 1976). With this choice, $s^{(k+1),\theta}(x)$ satisfies property (#) by Proposition 1 in Appendix C, and the transition density of the reverse process is SE(3)-invariant by Proposition 2 in Appendix C. Since the prior $p(x^{(N)})$ is also SE(3)-invariant, we conclude that the learned distribution $p_\theta(x^{(0)})$ is SE(3)-invariant (Xu et al., 2022). Compared to the commonly used equivariant networks (Satorras et al., 2021), our network fits our experiment better thanks to its lower computational cost and reduced memory usage.

For the Newton's method, we set $\mathrm{tol} = 10^{-5}$ and $n_{\mathrm{step}} = 10$ in Algorithm 4. For all but one point, the convergence is reached within 2 steps.

## C   THEORETICAL RESULTS ON NEURAL NETWORKS FOR MOLECULAR SYSTEMS

In this section, we present theoretical results for the neural network architecture we employed in studying alanine dipeptide.

Assume that the system consists of $M$ atoms, where the coordinates of thee $i$-th atom are denoted by $\boldsymbol{x}_i \in \mathbb{R}^3$, for $i = 1, 2, \ldots, M$. Let $x \in \mathbb{R}^{3M}$ be the vector consisting of all the coordinates $\boldsymbol{x}_1 \, \boldsymbol{x}_2, \ldots, \boldsymbol{x}_M \in \mathbb{R}^3$. For simplicity, given a rotation matrix $R \in \mathrm{SO}(3)$ and a translation vector $w \in \mathbb{R}^3$, we use the conventional notation $Rx + w$ to denote the vector in $\mathbb{R}^{3M}$ that consists of the transformed coordinates $R\boldsymbol{x}_1 + w, R\boldsymbol{x}_2 + w, \ldots, R\boldsymbol{x}_M + w \in \mathbb{R}^3$. We say that a function $f$ defined in $\mathbb{R}^{3M}$ is SE(3)-invariant, if $f(Rx + w) = f(x)$, for all $R \in \mathrm{SO}(3)$, $w \in \mathbb{R}^3$, and for all $x \in \mathbb{R}^{3M}$. We say that function $f : \mathbb{R}^{3M} \to \mathbb{R}^{3M}$ possesses property (#), if it is both equivariant under rotations and invariant under translations, i.e.

$$f(Rx + w) = Rf(x), \text{ for all } R \in \mathrm{SO}(3), w \in \mathbb{R}^3, \text{and all } x \in \mathbb{R}^{3M}. \tag{#}$$

Assume that a configuration $x^{\mathrm{ref}}$ is chosen as reference. Given $x$, the optimal rotation matrix and the optimal translation vector, which minimize the RMSD

$$\mathrm{RMSD}(x; x^{\mathrm{ref}}) = \left( \frac{1}{M} |R(x - w) - x^{\mathrm{ref}}|^2 \right)^{\frac{1}{2}} \tag{56}$$

from the reference $x^{\mathrm{ref}}$, are denoted by $R_x^*$ and $w_x^*$, respectively.

The following result characterizes functions that are both equivariant under rotations and invariant under translations.

**Proposition 1.** *The following two claims are equivalent.*

- *Function $s : \mathbb{R}^{3M} \to \mathbb{R}^{3M}$ possesses property (#).*

- *There is a function $f : \mathbb{R}^{3M} \to \mathbb{R}^{3M}$, such that $s(x) = (R_x^*)^\top f(R_x^*(x - w_x^*))$, for all $x \in \mathbb{R}^{3M}$.*

*Proof.* It is straightforward to verify that the first claim implies the second claim. In fact, setting $R = R_x^*$, $w = -R_x^* w_x^*$, and using the identity $R^\top R = I_3$, we obtain from the first claim that $s(x) = (R_x^*)^\top s(R_x^*(x - w_x^*))$. Hence, the second claim holds with $f = s$. To show that the second claim also implies the first one, we use the fact that the optimal rotation $R_{Rx+w}^*$ and the optimal translation $w_{Rx+w}^*$, which minimize the RMSD of the state $Rx + w$ from the reference $x^{\mathrm{ref}}$, are given by $R_{Rx+w}^* = R_x^* R^\top$ and $w_{Rx+w}^* = Rw_x^* + w$, respectively. This fact can be directly checked using equation 56. In particular, we have

$$R_{Rx+w}^*(Rx + w - w_{Rx+w}^*) = R_x^*(x - w_x^*).$$

Therefore, for the function $s$ defined in the second claim, we can compute, for any $R \in \mathrm{SO}(3)$, $w \in \mathbb{R}^3$, and any $x \in \mathbb{R}^{3M}$,

$$\begin{aligned} s(Rx + w) &= (R_{Rx+w}^*)^\top f\big(R_{Rx+w}^*(Rx + w - w_{Rx+w}^*)\big) \\ &= R(R_x^*)^\top f(R_x^*(x - w_x^*)) \\ &= Rs(x) \,, \end{aligned}$$

which shows the first claim. $\qquad\square$

The following result guarantees the SE(3)-invariance of the transition densities of our diffusion model.

**Proposition 2.** *Assume that $\xi$ is* SE(3)*-invariant and $b$ possesses property (#). Then, the transition density of the forward process in equation 10 is* SE(3)*-invariant. Further assume that the function $s^{(k+1),\theta}$ possesses property (#) for $0 \leq k \leq N - 1$. Then, the transition density of the reverse process in equation 12 is also* SE(3)*-invariant.*

*Proof.* We consider the transition density in equation 10. Recall that $U_x \in \mathbb{R}^{n \times d}$ is a matrix whose columns form an orthonormal basis of $T_x\mathcal{M}$. Since $\xi$ is SE(3)-invariant, we have $\xi(Rx+w) = \xi(x)$, for all rotations $R$ and translation vectors $w$, which implies that $Rx + w \in \mathcal{M}$, if and only if $x \in \mathcal{M}$. Differentiating the identity $\xi(Rx + w) = \xi(x)$ with respect to $x$, we obtain the relation $\nabla\xi(Rx + w) = R\nabla\xi(x)$, from which we see that $U_{Rx+w}$ can be chosen such that $U_{Rx+w} = RU_x$. For the orthogonal projection matrix $P$ in equation 19, using the identity $R^\top R = I_3$, we can compute

$$
\begin{aligned}
P(Rx + w) &= I_n - \nabla\xi(Rx + w)\big(\nabla\xi(Rx + w)^\top \nabla\xi(Rx + w)\big)^{-1}\nabla\xi(Rx + w)^\top \\
&= I_n - R\nabla\xi(x)\big(\nabla\xi(x)^\top \nabla\xi(x)\big)^{-1}\nabla\xi(x)^\top R^\top \\
&= RP(x)R^\top .
\end{aligned}
$$

Moreover, since both $b$ and $\nabla\xi$ satisfy the property (#), we also have $\epsilon(Rx^{(k)}+w; \sigma_k) = \epsilon(x^{(k)}; \sigma_k)$ (i.e. the probabilities of having no solution are the same). Therefore, for the transition density in equation 10, we can derive, for any $R \in$ SO(3) and $w \in \mathbb{R}^3$,

$$
q(Rx^{(k+1)} + w \,|\, Rx^{(k)} + w)
$$
$$
=(2\pi\sigma_k^2)^{-\frac{d}{2}}\big(1 - \epsilon(Rx^{(k)} + w; \sigma_k)\big)^{-1}|\det(U_{Rx^{(k)}+w}^\top U_{Rx^{(k+1)}+w})|
$$
$$
\times \mathrm{e}^{-\frac{\left|P(Rx^{(k)}+b)\left(Rx^{(k+1)} - Rx^{(k)} - \sigma_k^2 b(Rx^{(k)}+w)\right)\right|^2}{2\sigma_k^2}}
$$
$$
=(2\pi\sigma_k^2)^{-\frac{d}{2}}\big(1 - \epsilon(x^{(k)}; \sigma_k)\big)^{-1}\left|\det(U_{x^{(k)}}^\top R^\top RU_{x^{(k+1)}})\right|\mathrm{e}^{-\frac{\left|RP(x^{(k)})R^\top\left(Rx^{(k+1)} - Rx^{(k)} - \sigma_k^2 Rb(x^{(k)})\right)\right|^2}{2\sigma_k^2}}
$$
$$
=(2\pi\sigma_k^2)^{-\frac{d}{2}}\big(1 - \epsilon(x^{(k)}; \sigma_k)\big)^{-1}\left|\det(U_{x^{(k)}}^\top U_{x^{(k+1)}})\right|\mathrm{e}^{-\frac{\left|P(x^{(k)})\left(x^{(k+1)} - x^{(k)} - \sigma_k^2 b(x^{(k)})\right)\right|^2}{2\sigma_k^2}}
$$
$$
=q(x^{(k+1)} \,|\, x^{(k)}) ,
$$

which shows the SE(3)-invariance of the transition density of the forward process. The invariance of the transition density of the reverse process in equation 12 can be proved using the same argument, assuming that $s^{(k+1),\theta}$ satisfies the relation $s^{(k+1),\theta}(Rx + w) = Rs^{(k+1),\theta}(x)$. $\square$

# D    SUPPLEMENTARY EXPERIMENTAL RESULTS

This section presents additional experimental results. In Section D.1, we present the computation time for the experiments in the main text. Section D.2 analyzes the computational complexity of our projection scheme and the non-convergence rate of trajectories. Finally, in Section D.3, we conduct an ablation study on the Flood dataset, investigating the impact of the total number of steps $N$ and the trajectory update frequency (proportional to $l_\mathrm{f}^{-1}$).

## D.1    COMPUTATION TIME FOR THE EXPERIMENTS IN THE MAIN TEXT

In Table 4, we present the simulation time $T_\mathrm{sim}$ and training time $T_\mathrm{train}$, with the percentages of the total runtime $T_\mathrm{total}$ shown in parentheses. For the mesh, SO(10), and the dipeptide datasets, we update the dataset every 100 epochs (i.e. $l_f = 100$), resulting in the simulation time accounting for less than 2% of the total runtime in mesh and dipeptide datasets, and the simulation time accountingfor approximately 11% of the total runtime in the SO(10) datasets due to the high co-dimension of the manifold. In contrast, for the earth and climate science datasets, the dataset is updated at every epoch (i.e. $l_f = 1$), leading to a higher proportion of simulation time.

Table 4: Detailed runtime metrics in our experiments. We report $T_{\text{sim}}$, $T_{\text{train}}$, and $T_{\text{total}}$ as the time for path generation, time for training, and total runtime, respectively, with the percentages of the total runtime $T_{\text{total}}$ shown in parentheses. The parameter $l_{\text{f}}$ determines the frequency of trajectory updates. The final column $T_{\text{epoch}}$ shows the training time per epoch, calculated as $T_{\text{train}}/N_{\text{epoch}}$. All time metrics are reported in seconds.

| Datasets | $l_{\text{f}}$ | $N_{\text{epoch}}$ | $T_{\text{sim}}$ | $T_{\text{train}}$ | $T_{\text{total}}$ | $T_{\text{epoch}}$ |
|---|---|---|---|---|---|---|
| Volcano | 1 | 20000 | 3946(67.0%) | 1943(33.0%) | 5889 | 0.10 |
| Earthquake | 1 | 20000 | 4591(24.2%) | 14395(75.8%) | 18986 | 0.72 |
| Flood | 1 | 20000 | 4414(29.0%) | 10814(71.0%) | 15228 | 0.54 |
| Fire | 1 | 20000 | 5666(14.0%) | 34893(86.0%) | 40559 | 1.74 |
| Bunny, $k = 50$ | 100 | 2000 | 252(1.8%) | 14120(98.2%) | 14372 | 7.06 |
| Bunny, $k = 100$ | 100 | 2000 | 142(1.6%) | 8718(98.4%) | 8860 | 4.36 |
| Spot, $k = 50$ | 100 | 2000 | 113(1.3%) | 8751(98.7%) | 8864 | 4.38 |
| Spot, $k = 100$ | 100 | 2000 | 70(1.3%) | 5189(98.7%) | 5259 | 2.59 |
| SO(10), $m = 3$ | 100 | 2000 | 2414(11.2%) | 19042(88.8%) | 21456 | 9.52 |
| SO(10), $m = 5$ | 100 | 2000 | 2426(11.2%) | 19289(88.8%) | 21715 | 9.64 |
| Alanine dipeptide | 100 | 5000 | 159(1.1%) | 14299(98.9%) | 14458 | 2.86 |

### D.2 MORE DETAILS ABOUT THE PROJECTION SCHEME

**Computational complexity of the projection scheme.** Let $k_{\text{iter}}$ denote the number of Newton iterations, and $C_\xi$ be the computational cost of evaluating $\nabla\xi$. The complexity of solving the linear equations in Algorithm 4 is $\mathcal{O}((n - d)^3)$ at maximum, where $n - d$ is the co-dimension of the manifold. Thus, the total complexity of Newton's method is $\mathcal{O}(k_{\text{iter}}(C_\xi + (n - d)^3))$.

In our examples, we have $C_\xi = \mathcal{O}(1)$, except for the mesh datasets, where neural network propagation is required. In most cases, $n - d = 1$, while for the SO(10) datasets, $n - d = 55$. When Newton's method converges, the number of iterations $k_{\text{iter}}$ is no more than 3.

**Failure rate of trajectory generation.** As discussed in Section 3.1, equation 5 may not have solutions for some vectors $v$. In such case, the corresponding trajectories are discarded. Table 5 reports the failure rate of trajectory generation in our experiments. For the earth and climate science datasets, the general scheme in equation 5 is computed analytically using equation 48, and no trajectories are discarded with parameters in Table 3. For the mesh dataset, due to the complex geometry of the manifolds, a small portion of trajectories (less than 1.0%) fail. For SO(10), trajectory generation always succeeds. In the alanine dipeptide experiment, only one out of 8000 trajectories fails.

Furthermore, we investigate the impact of the step-size $\sigma$ on the failure rate of Newton's method. Specifically, as shown in Table 6, we conduct experiments on the Bunny dataset with $k = 100$, using various values of $N$, where $N$ and $\sigma$ are related by $\sigma \sim 1/\sqrt{N}$. As the total number of steps $N$ decreases (i.e., as the step-size $\sigma$ increases), the proportion of failed trajectories increases.

### D.3 ABLATION STUDIES

In Table 7, we investigate the impact of the total number of steps $N$ in the Markov chain on the runtime for the Flood dataset. In Table 8, we analyze the effect of the trajectory update frequency during training on the Flood dataset.

As shown in Table 7, as the total number of steps $N$ decreases (i.e., as the step-size $\sigma$ increases), the Newton-based projection scheme may fail to find a solution in some cases. Specifically, when $N = 50$ and $N = 100$, the failure rates are 0.55% and 0.0001%, respectively. When $N = 200$ and $N = 400$, no failure is observed. Additionally, as $N$ decreases, both the training time and the simulation time reduce accordingly. The best NNL value is achieved when $N = 100$. However, in the main text, we point out the issue of NNL value in this example and choose $N = 400$ to ensure

Table 5: Failure rate of trajectory generation. $R_{\text{fail\_fwd}}$ and $R_{\text{fail\_bwd}}$ represent the percentages of discarded trajectories when sampling the forward and reverse process, respectively. $\sigma_{\text{max}}$ denotes the maximum value of $(\sigma_k)_{0 \leq k \leq N-1}$.

| Datasets | $\sigma_{\text{max}}$ | $R_{\text{fail\_fwd}}$ | $R_{\text{fail\_bwd}}$ |
|---|---|---|---|
| Volcano | 0.100 | 0.00% | 0.00% |
| Earthquake | 0.100 | 0.00% | 0.00% |
| Flood | 0.100 | 0.00% | 0.00% |
| Fire | 0.100 | 0.00% | 0.00% |
| Bunny, $k = 50$ | 0.007 | 1.00% | 0.82% |
| Bunny, $k = 100$ | 0.007 | 0.65% | 0.55% |
| Spot, $k = 50$ | 0.010 | 0.15% | 0.25% |
| Spot, $k = 100$ | 0.010 | 0.11% | 0.10% |
| SO(10), $m = 3$ | 0.089 | 0.00% | 0.00% |
| SO(10), $m = 5$ | 0.089 | 0.00% | 0.00% |
| Alanine dipeptide | 0.022 | 0.01% | 0.00% |

Table 6: Failure rate of trajectory generation for the Bunny dataset with $k = 100$ under various $N$. Here, $\sigma$ denotes the step-size and $R_{\text{fail\_fwd}}$ represents the proportion of failed forward trajectories.

| $N$ | 100 | 200 | 300 | 400 | 500 | 600 | 700 |
|---|---|---|---|---|---|---|---|
| $\sigma$ | 1.57e-2 | 1.11e-2 | 0.90e-2 | 0.78e-2 | 0.70e-2 | 0.64e-2 | 0.59e-2 |
| $R_{\text{fail\_fwd}}$ | 3.51% | 2.30% | 1.49% | 0.97% | 0.66% | 0.46% | 0.17% |

that all trajectories are successfully generated. Furthermore, as shown in Table 8, increasing the frequency of trajectory updates (decreasing $l_f$) leads to longer computational time.

Table 7: Ablation study on the total number of steps $N$ in the Markov chain for the Flood dataset. The parameter $R_{\text{fail\_fwd}}$ denotes the percentage of discarded trajectories when simulating the forward process. The definitions of $T_{\text{sim}}, T_{\text{train}}, T_{\text{total}}, T_{\text{epoch}}$ can be found in the caption of Table 4.

| $N$ | $\sigma_{\text{max}}$ | $R_{\text{fail\_fwd}}$ | $T_{\text{sim}}$ | $T_{\text{train}}$ | $T_{\text{total}}$ | $T_{\text{epoch}}$ | NLL |
|---|---|---|---|---|---|---|---|
| 50 | 0.28 | 0.55% | 643(30.8%) | 1442(69.2%) | 2085 | 0.07 | $0.48_{\pm 0.08}$ |
| 100 | 0.20 | 0.00% | 1151(29.1%) | 2801(70.9%) | 3952 | 0.14 | $0.45_{\pm 0.07}$ |
| 200 | 0.14 | 0.00% | 2245(29.0%) | 5497(71.0%) | 7742 | 0.27 | $0.46_{\pm 0.07}$ |
| 400 | 0.10 | 0.00% | 4414(29.0%) | 10814(71.0%) | 15228 | 0.54 | $0.49_{\pm 0.09}$ |

Table 8: Ablation study on the trajectory update frequency (proportional to $l_{\mathrm{f}}^{-1}$) for the Flood dataset with $N = 100$. The definitions of $R_{\mathrm{non\_c}}, T_{\mathrm{sim}}, T_{\mathrm{train}}, T_{\mathrm{total}}, T_{\mathrm{epoch}}$ can be found in the caption of Table 4.

| $l_{\mathrm{f}}$ | $T_{\mathrm{sim}}$ | $T_{\mathrm{train}}$ | $T_{\mathrm{total}}$ | $T_{\mathrm{epoch}}$ | NLL |
|---|---|---|---|---|---|
| 1 | 1151(29.1%) | 2801(70.9%) | 3952 | 0.14 | $0.45_{\pm 0.07}$ |
| 2 | 556(16.6%) | 2792(83.4%) | 3348 | 0.14 | $0.49_{\pm 0.09}$ |
| 3 | 366(11.4%) | 2838(88.6%) | 3204 | 0.14 | $0.49_{\pm 0.08}$ |
| 5 | 230(7.6%) | 2787(92.4%) | 3017 | 0.14 | $0.48_{\pm 0.09}$ |
| 10 | 111(2.9%) | 3682(97.1%) | 3793 | 0.18 | $0.45_{\pm 0.07}$ |

