# OpenReview forum: "Riemannian denoising diffusion probabilistic models"
_ICLR.cc/2025/Conference — Submitted to ICLR 2025_

### Official Review · Reviewer_fyyt · 2024-10-26

**Soundness:** 3
**Presentation:** 2
**Contribution:** 2
**Rating:** 6
**Confidence:** 2

**Summary:**

This paper presents Riemannian Denoising Diffusion Probabilistic Models (RDDPMs), a denoising diffusion model for learning distributions over submanifolds of Euclidean space. The key insight stems from the fact that a projection-based approach becomes relatively straightforward when the manifold is represented as the zero-level set of some smooth function $\xi$. Unlike previous work, the method proposed here does not require sampling of geodesic paths, the heat kernel on the manifold, or other similar geometric quantities. The method uses the fact that forward and reverse process transition densities for the underlying diffusion Markov chain become tractable if one has access to $\nabla \xi$. The authors present an algorithm to sample trajectories from the forward and reverse process using Newton's Method. They show that in the limit of infinitely many denoising steps, the objective converges to the continuous time score matching objective from previous work.

**Strengths:**

(1) The projection-based method is easy to understand, and doesn't require the exponential map or the manifold's heat kernel.

(2) The approach is well-motivated, as previous work either (a) relies on substantial geometric information about the manifold that isn't typically available, or (b) assumes a restricted class of manifolds (i.e. symmetric manifolds)

(3) A theoretical connection to continuous-time score based generative modeling is established in the limit of infinite resolution timesteps.

**Weaknesses:**

(1) One claim about previous work seems unsubstantiated: In line 053, the paper states that existing continuous-time methods suffer from error based on time-discretization, something that RDDPMs avoid. But no experiments are provided to justify this. Furthermore, results in Table 1 suggest that RDDPM does not, in general, perform better than existing approaches.

(2) Some of the presentation is a little unclear:
- In line 137, the function $b: \mathbb{R}^n \rightarrow \mathbb{R}^n$ is referenced, but not defined or discussed until line 235. Can some of this discussion be moved to line 137?
- In line 138, $\nabla \xi$ is referred to as an orthogonal direction, even though it is a matrix $\in \mathbb{R}^{n \times (n-d)}$. Should the wording perhaps be "along an orthogonal direction in the column space of $\nabla \xi$?
- In Table 1, were the baselines trained with isolated points as well?
- Can the best-performing approaches in Tabe 1 be underlined or starred? This would make it easier to parse.

(3) Poor RDDPM performance on real data (Table 1): the real data evaluations indicate that RDDPM performs worse than baseline methods. Furthermore, no discussion of these results is provided.

(4) The method seems extremely computationally expensive. Generating forward or reverse process trajectories requires $N$ calls to Newton's method. Furthermore, if Newton's method doesn't find a projection, the entire trajectory is discarded.

(5) The paper doesn't acknowledge the case when the projection operator is not unique (which can occur if the step size is large enough in comparison to the reach of the manifold). Can some discussion about the potential consequences of this be added?

(6) The projection scheme isn't a true orthogonal or closest point projection, as $\nabla\xi(x) \neq \nabla\xi(y)$ in general. While in the continuous-time limit this doesn't seem to be a problem, I would appreciate some more discussion about this.

**Questions:**

(1) Do all d-dimensional smooth, compact, and connected submanifolds of $\mathbb{R}^n$ fit within this framework?

(2) Do the columns of $\xi(x), x \in \mathcal{M}$ form a basis of the normal space to $\mathcal{M}$ at $x$?

(3) Empirically, how often does Newton's method fail (and the trajectory needs to be discarded)? This question applies to sampling both forward and reverse process trajectories.

(4) Can existing approaches be evaluated for the SO(10) group or Alanine Dipeptide experiments for comparison? If not, can the reasons be described?

---

> ### Author Response · Authors · 2024-11-20
> **Author Response to Reviewer fyyt (part 1)**
>
> We are grateful to the reviewer for the careful review and constructive comments.
>
> ## - Weaknesses assessed by the reviewer
>
> 1. One claim about previous work seems unsubstantiated: In line 053, the paper states that existing continuous-time methods suffer from error based on time-discretization, something that RDDPMs avoid. But no experiments are provided to justify this. Furthermore, results in Table 1 suggest that RDDPM does not, in general, perform better than existing approaches.
>
>     **Reply**: In line 53, we intended to discuss the advantage to have a theoretical framework that avoids error due to time-discretization. Following the reviewer's suggestion, we have removed that sentence in the revised manuscript. Table 1 shows the results for data on 2D sphere. Because this is a very special/simple manifold with analytical geodesics and heat kernel, we think it is not representative for comparing our method to existing methods. We have studied this example and discuss issues related to these datasets (with numerical support in Appendix B.1), which we believe is helpful to further studies on this example.
>
> 2. In line 137, the function $b:\mathbb{R}^n\rightarrow\mathbb{R}^n$ is referenced, but not defined or discussed until line 235. Can some of this discussion be moved to line 137?
>
>    **Reply**: Thank you for your suggestion. We have made the corresponding revision in our revised manuscript.
>
> 3. In line 138, $\nabla\xi$ is referred to as an orthogonal direction, even though it is a matrix $\in \mathbb{R}^{n\times(n-d)}$. Should the wording perhaps be "along an orthogonal direction in the column space of $\nabla\xi$?
>
>    **Reply**: We agree. We have revised that sentence in the revised manuscript.
>
> 4. In Table 1, were the baselines trained with isolated points as well?
>
>    **Reply**: To our knowledge, this phenomenon (the existence of isolated points and its consequence) associated to the datasets has not been mentioned in the previous works.
>
> 5. Can the best-performing approaches in Table 1 be underlined or starred? This would make it easier to parse.
>
>    **Reply**: By presenting a detailed discussion in this example on the issue related to the datasets, we hope to present readers a comprehensive view on the matter rather than only NNL scores. In consistent with this, we prefer not to highlight numbers in this table.
>
> 6. Poor RDDPM performance on real data (Table 1): the real data evaluations indicate that RDDPM performs worse than baseline methods. Furthermore, no discussion of these results is provided.
>
>    **Reply**:  Table 1 shows results for data on 2D sphere. This is one of a few special manifolds with analytical geodesics and heat kernel to which existing methods can be easily applied. In fact, we have provided the following discussions (Lines 390 to 397) on the issue related to comparing different methods on these datasets.
>
>    > ``*We point out a general issue regarding model evaluation on these datasets. We notice that in each dataset there are a few data points (i.e. isolated points) whose vicinity contains no other points, and the size of dataset is relatively small compared to its complex distribution. For such datasets, standard dataset splitting for cross-validation results in non-overlapping training/validation/test sets, whose empirical distributions are considerably different. As a consequence, computing the NLL on the test set either results in an overconfident assessment of the model (when the test set contains no isolated points) or requires evaluating the model on isolated points that are completely unseen during training (when the test set contains isolated points).*''
>
>    This issue should exist in previous studies but it has not been discussed. We believe our discussion is helpful for further studies on these datasets.

---

> ### Author Response · Authors · 2024-11-20
> **Author Response to Reviewer fyyt (part 2)**
>
> ## - Weaknesses assessed by the reviewer
>
> 7. The method seems extremely computationally expensive. Generating forward or reverse process trajectories requires $N$ calls to Newton's method. Furthermore, if Newton's method doesn't find a projection, the entire trajectory is discarded.
>
>    **Reply**: In fact, the method is not that expensive. In section 3.5, we have the following discussion on Newton's method.
>
>    >  ``*This method has quadratic convergence (locally) and its implementation is simple. In most cases, a solution with high precision can be found within a few iteration steps (e.g. less than 5 steps). When no solution is found, one can re-generate the state or the entire trajectory.*''
>
>    In our numerical experiment, with properly chosen step-size, only in the second example (mesh data) where less than $1.0\%$ of trajectories can not be generated due to the curved shape of the geometric objects. In other three examples, trajectory generation is always successful except one or two cases. The Newton's method finds a solution typically within 3 steps. While in our implementation the trajectory is simply discarded in case of no solution, one can resample a tangent vector $v$ and try to re-generate a new state.  Please refer to our ``Reply to common questions from reviewers'' above and Tables 4-6 in Appendix D in the revised manuscript for detailed discussions on the computational cost.
>
> 8. The paper doesn't acknowledge the case when the projection operator is not unique (which can occur if the step size is large enough in comparison to the reach of the manifold). Can some discussion about the potential consequences of this be added?
>
>    **Reply**: We admit that in our initial submission the discussion on the non-uniqueness of the solutions to the constraint equation was not clear. In fact, there may be multiple solutions to the constraint equation, but this is not an issue because we are considering a Markov chain determined by a selection scheme (i.e. a scheme to select one solution in case of multiple solutions). In practice, such scheme could be Newton's method with fixed initial condition. We have revised our manuscript to clarify this point. Please refer to our ``Reply to common questions from reviewers'' above for detailed discussions on this point.
>
> 9. The projection scheme isn't a true orthogonal or closest point projection, as $\nabla \xi(x)\neq \nabla \xi(y)$  in general. While in the continuous-time limit this doesn't seem to be a problem, I would appreciate some more discussion about this.
>
>    **Reply**: Yes, the projection scheme employed in our algorithm is not an orthogonal or closest point projection in traditional sense. However, we think that this is a feature of our proposed method rather than a weakness. In fact, it allows us to obtain an explicit transition probability density in Equation (8), which in turn leads to a computable training objective in Equation (14). In Section 5, we have discussed our projection scheme compared to the closest point projection (Lines 357-362).
>
> ## - Questions raised by the reviewer
>
> 1. Do all d-dimensional smooth, compact, and connected submanifolds of $\mathbb{R}^n$ fit within this framework?
>
>    **Reply**: Yes, our framework can be applied to these general submanifolds of $\mathbb{R}^n$, as long as they can be defined as the level set of certain function $\xi$.
>
> 2. Do the columns of $\xi(x)$, $x\in \mathcal{M}$ form a basis of the normal space to $\mathcal{M}$ at $x$?
>
>    **Reply**: We believe that the reviewer meant $\nabla\xi(x)$ instead of $\xi(x)$. Yes, the columns of $\nabla\xi(x)$ form a basis of the normal space to $\mathcal{M}$ (space orthogonal to the tangent space) at $x$.
>
> 3. Empirically, how often does Newton's method fail (and the trajectory needs to be discarded)? This question applies to sampling both forward and reverse process trajectories.
>
>    **Reply**: As discussed in our reply to a previous comment from the reviewer, in our numerical experiment, only in the second example (mesh data) where less than $1.0\%$ of trajectories can not be generated due to the curved shape of the geometric objects. In other three examples, trajectory generation is always successful except one or two cases. Similar is true for the reverse process. While the trajectory is simply discarded in our implementation, one can resample a tangent vector $v$ and try to re-generate a new state.
>
>    In our initial submission, we have provided discussions on Newton's method in the main text and given these empirical details in Appendix B.1-B.3. In the revised manuscript, we summarize them in Tables 5-6 in Appendix D, so that they can be easily found.

---

> ### Author Response · Authors · 2024-11-20
> **Author Response to Reviewer fyyt (part 3)**
>
> ## - Questions raised by the reviewer
>
> 4. Can existing approaches be evaluated for the SO(10) group or Alanine Dipeptide experiments for comparison? If not, can the reasons be described?
>
>    **Reply**:  Existing methods require geometric information of the manifold, such as geodesics, heat kernel, eigenfunctions of Laplace-Beltrami, or a mesh grid of the manifold. For Alanine Dipeptide with fixed dihedral angle, existing methods are not applicable because the manifold is implicitly defined and none of these information is available. For special orthogonal groups SO(n), current works are limited to orthogonal groups of order $n=3$ and higher order groups such as SO(10) have not been studied. This limitation, in our opinion, is due to the computational difficulty of the geometric information such as the exponential map and heat kernel for higher-dimensional orthogonal groups. In contrast, our approach circumvents the need to compute these quantities and relies solely on a projection framework.

---

> > ### Comment · Reviewer_fyyt · 2024-11-24
> > **Response to rebuttal**
> >
> > Thank you for the detailed response!
> >
> > *Yes, our framework can be applied to these general submanifolds of $\mathbb{R}^n$, as long as they can be defined as the level set of certain function $\xi$.*
> >
> > It would be nice if some discussion about what kind of manifolds can be defined as the level set of certain functions. I imagine that with no restriction on the function class, all manifolds can be represented in this way (but I am no authority on the matter). Some discussion here would go a long way I believe.
> >
> > *In other three examples, trajectory generation is always successful except one or two cases.*
> >
> > One or two cases fail out of how many attempted trajectory generations?

---

> > > ### Author Response · Authors · 2024-11-26
> > >
> > > We thank the reviewer for the further questions.
> > >
> > > Regarding the first question, we think that it is correct. In fact, a general compact Riemmanian submanifold $\mathcal{M}$ can always be defined as the zero level set of the distance function $f(x)= d(x,\mathcal{M})$, although this definition is possibly not useful because it involves the manifold itself. It is also possible that a manifold defined as a level set has a more direct/explicit representation. For these reasons, we do not claim that all submanifolds fit into this setting or should be studied in this way, but we only say that it covers a large class of manifolds in applications in molecular dynamics and statistical mechanics (lines 48-49 in the revised manuscript). We are particularly motivated by applications in molecular dynamics where the distribution of the system under fixed collective variables, e.g. angles or bond lengths, is of interest (the alanine dipeptide example in our paper). We are willing to consider to add a short discussion or one or two references in the final version of our manuscript.
> > >
> > > Regarding the second question, sorry for being imprecise in our previous response. In fact, we have provided the details in Appendix D of our revised manuscript (in lines 1491-1495):
> > >
> > > > ``*For the earth and climate science datasets, the general scheme in equation 5 is computed analytically using equation 48, and no trajectories are discarded with parameters in Table 3... For SO(10), trajectory generation always succeeds. In the alanine dipeptide experiment, only one out of 8000 trajectories fails.*''

---

> > > > ### Comment · Reviewer_fyyt · 2024-11-26
> > > > **Response to rebuttal**
> > > >
> > > > Thank you for addressing my questions. While the authors have diligently responded to my comments and concerns, I will keep my score as is.

---

### Official Review · Reviewer_gqWG · 2024-10-31

**Soundness:** 3
**Presentation:** 3
**Contribution:** 2
**Rating:** 5
**Confidence:** 3

**Summary:**

This paper introduces a method for generative modelling when the data is specified on a pre-defined Riemannian submanifold of Euclidean space. In contrast to previous methods, it uses a projection scheme to construct trajectories on the manifold, only requiring gradient information of the defining equation for the manifold. Theoretical derivations show connections between the continuous-time limit of this method and Riemannian score-based methods. The projections onto the manifold for the forward and reverse process are computed by solving a system of equations using Newton's method. This method is then tested on datasets supported on different manifolds.

**Strengths:**

The techniques in this paper allows for training diffusion models on a more general class of manifolds than other methods, only requiring knowing the defining equation of the manifold. It solves a system of equations to ensure the points generated by the forward and reverse processes stays on the manifold, instead of requiring knowledge of the closest-point projection onto the manifold. There are extensive experiments on a variety of manifolds and datasets.

**Weaknesses:**

In my opinion, the disadvantages of this method (outlined below) outweighs the main advantages (allowing for more general manifolds). The main contribution seems to be solving for the projection numerically, which is only useful when the exact projection onto the manifold is not known, but there needs to be more evaluation on this front.

1. During training, the entire trajectory for the random walk needs to be generated.  The training objective predicts the expected value of $x^{k}$ given $x^{k+1}$, unlike score matching which predicts the expected value of $x^{0}$ given $x^{k+1}$.
2. During sampling, because trajectory generation can fail, small steps are necessary as taking large steps will increase the chances of (11) having many or no solutions.
3. Because this method predicts $x^{k}$ given $x^{k+1}$, the number of sampling steps need to be the same as that of training steps (200-800 steps). This is large compared to <50 steps needed for fast samplers of diffusion models in Euclidean space.
4. The analysis in the paper does not consider cases where the projection is not unique, or when equations (9) and (11) have no solutions.
5. The evaluation for higher dimension manifolds only compares histograms of some statistics of the forward process and reverse process, and it is unclear from these how close the generated distributions are to each other.

**Questions:**

I would suggest that the authors try this method on more manifolds where only the defining equations are known but without closed-form projections, and devise better evaluation metrics for these manifolds.

Questions:
1. Have the authors considered using diffusion models in Euclidean space to learn the distribution on the manifold? The generated distribution may not lie exactly on the manifold, but one can project onto the manifold as a final step after diffusion sampling.

2. Is there a better way (e.g. total variation distance) to evaluate the generated distributions on SO(10) and alanine dipeptide where one can compare numerical values instead of histograms?

3. It would also be nice to have some ablation studies on the hyperparameters of the experiments.

---

> ### Author Response · Authors · 2024-11-20
> **Author Response to Reviewer gqWG (part 1)**
>
> We sincerely thank the reviewer for the thoughtful comments and valuable recommendations.
>
> ## - Weaknesses assessed by the reviewer
>
> 1. In my opinion, the disadvantages of this method (outlined below) outweighs the main advantages (allowing for more general manifolds). The main contribution seems to be solving for the projection numerically, which is only useful when the exact projection onto the manifold is not known, but there needs to be more evaluation on this front.
>
>    **Reply**: We believe that the main contribution of our work is more than ``solving for the projection numerically...''. In fact, we have proposed a generative modeling approach in a general setting where the manifold is defined as a level-set of certain function. We have mentioned that this setting fits well with applications in molecular dynamics and statistical mechanics where constraints are often involved (Lines 48-50), and these applications can not be handled by existing methods that require heat kernel or eigenfunctions of Laplace-Beltrami operator. We have applied our approach to a concrete example (molecular system alanine dipeptide) in this setting. We believe that the exploration of this new scenario is both meaningful and valuable. We have also refined our analysis such that the cases where the constraint equation has multiple solutions or no solution do not cause any theoretical issue.
>
> 2. During training, the entire trajectory for the random walk needs to be generated. The training objective predicts the expected value of $x^k$ given $x^{k+1}$, unlike score matching which predicts the expected value of $x^0$ given $x^{k+1}$.
>
>    **Reply**: Concerning the first comment, it is true that the entire trajectory needs to be generated. However, all pairs ($x^{(k)}$, $x^{(k+1)}$) at different steps $k$ along the trajectory are useful as they all contribute to the training objective. In Section 3.5, we mention that generating trajectories can be done in a preparatory step before training, and the trajectory dataset can be updated every several training epochs.
>
>    Concerning the second comment, we think that it is only true in Euclidean space. For score-based diffusion models on manifolds, one needs not only to simulate the geodesic random walk but also know the transition kernel, both of which are not easy except for a few special manifolds. Please also refer to our ``Reply to common questions from reviewers'' above for discussions on trajectory simulations.
>
> 3. During sampling, because trajectory generation can fail, small steps are necessary as taking large steps will increase the chances of (11) having many or no solutions.
>
>    **Reply**: We have refined our analysis to take into account the cases where there are multiple solutions or no solution. Please refer to our ``Reply to common questions from reviewers'' for discussions on the choice of step-size.
>
> 4. Because this method predicts $x^k$ given $x^{k+1}$, the number of sampling steps need to be the same as that of training steps (200-800 steps). This is large compared to $<$50 steps needed for fast samplers of diffusion models in Euclidean space.
>
>    **Reply**: While for concrete applications $<$50 steps may be enough, we think that in general the proper number of sampling steps is problem dependent and direct comparison of this number among different applications may not be appropriate. In fact, the number of sampling steps affects the step-size of the sampling scheme used to simulate the learned diffusion model, and it is known from the theory of numerical analysis that the theoretical error bound of numerical schemes for sampling SDEs (or ODEs) depends on the coefficients of the SDEs and the error bound increases as the step-size of the numerical scheme increases. From this perspective, we think that in general the number of sampling steps should be chosen (taking into account the magnitude of the learned score) such that the learned diffusion model is well approximated (according to certain criteria).
>
>    In this work we consider generating distributions on manifolds, which is inherently a more challenging problem compared to the Euclidean setting. In our experiments, we were conservative when choosing the number of discretization steps, since the main focus is to validate a new method rather than to optimize the parameters in specific applications.
>
> 5. The analysis in the paper does not consider cases where the projection is not unique, or when equations (9) and (11) have no solutions.
>
>    **Reply**: We admit that there were some issues in our presentation concerning the cases where the constraint equation has no solution or multiple solutions. We have refined the analysis and those issues are resolved in our revised manuscript. Please refer to our ``Reply to common questions from reviewers'' for detailed discussions on this point.

---

> ### Author Response · Authors · 2024-11-20
> **Author Response to Reviewer gqWG (part 2)**
>
> ## - Weaknesses assessed by the reviewer
>
> 6. The evaluation for higher dimension manifolds only compares histograms of some statistics of the forward process and reverse process, and it is unclear from these how close the generated distributions are to each other.
>
>    **Reply**: In the SO(10) and alanine dipeptide examples, the histograms show that the distributions (of statistics) of the forward and the reverse processes match well with each other at different intermediate steps $k$ (not only the generated data at $k=0$). While we can not exclude the case where the statistics coincidentally matches but the data does not, these statistics are not trivial and we think that their closeness provides evidence on the quality of the learned model.
>
> ## - Questions raised by the reviewer
>
> 1. I would suggest that the authors try this method on more manifolds where only the defining equations are known but without closed-form projections, and devise better evaluation metrics for these manifolds.
>
>    **Reply**: We have applied our method to study datasets from SO(10) and alanine dipeptide, where the manifolds are implicitly defined as level-sets of the defining equations. In Appendix B.3 and B.4, we have provided sufficient details about our numerical experiments on these two numerical examples. We believe these numerical studies, together with the theoretical analysis in Appendix A, are sufficient to demonstrate the capability of our method. The field of molecular dynamics provides many interesting applications, but due to the constraint of time and effort, we think that it is appropriate to leave them for future studies.  We hope that the reviewer can agree with our consideration.
>
> 2. Have the authors considered using diffusion models in Euclidean space to learn the distribution on the manifold? The generated distribution may not lie exactly on the manifold, but one can project onto the manifold as a final step after diffusion sampling.
>
>    **Reply**: We thank the reviewer for raising this natural question. In fact, we had a similar idea when we started working on this research project. The diffusion models in Euclidean space is a general approach for learning distributions from data and it may be effective when the manifold structure is simple. The issue is, however, that the score $\nabla\log q_t$ at $t=0$ is singular (its orthogonal component is infinity) because the target density $q_0$ is only defined in a low-dimensional subspace. In theory this could result in numerical issues in model training and in sampling (e.g. step-size in sampling needs to shrink to zero as $t$ gets closer to zero). It may be possible to adjust parameters to make things work in concrete applications  (as suggested by the reviewer, by stopping at certain small time $t_0$ and projecting from there), but we believe that it is meaningful to develop methods on manifolds with sound theoretical support.
>
> 3. Is there a better way (e.g. total variation distance) to evaluate the generated distributions on SO(10) and alanine dipeptide where one can compare numerical values instead of histograms?
>
>    **Reply**: We have thought about the reviewer's suggestion carefully. We noticed that Maximum Mean Discrepancy (MMD) and Wasserstein distances are adopted in literature for this purpose. Based on our empirical tests on these metrics, we noticed that it is crucial to use a large sample size in order to eliminate statistical fluctuations in computing these metrics. However, because in our case the data lives on a low-dimensional submanifold of the Euclidean space, these metrics can not be directly applied and it is necessary to make modifications (e.g. using manifold distance instead of Euclidean distance in computing Wasserstein distances), which often result in huge computational cost in computing these metrics when the sample size of (high-dimensional) data is large. With these considerations, we decide not to explore these alternatives because a good understanding is still lacking.
>
> 4. It would also be nice to have some ablation studies on the hyperparameters of the experiments.
>
>    **Reply**: Following the reviewer's suggestion, we have selected one numerical example and conducted ablation studies on some hyperparameters used there. The results are included in Appendix D.3 in our revised manuscript.

---

> ### Comment · Reviewer_gqWG · 2024-11-25
>
> Thank you for the detailed responses, as well as the updates to the paper. I remain unconvinced that the benefits of the approach in this paper outweigh the added complexity, when compared to diffusion models in Euclidean space. Denoising score matching is designed to learn low-dimensional data manifolds [1], and experiments on the swissroll dataset [2] show that Euclidean space diffusion models can also learn distributions on low-dimensional manifolds. Thus I will keep my score.
>
> [1] Song, Yang, and Stefano Ermon. "Generative modeling by estimating gradients of the data distribution."
>
> [2] Sohl-Dickstein, Jascha, et al. "Deep unsupervised learning using nonequilibrium thermodynamics."

---

> > ### Author Response · Authors · 2024-11-26
> >
> > We thank the reviewer for the response and for mentioning the works [1,2] on diffusion models in Euclidean space. As mentioned in our reply to the review's question, it can be shown that the score $\nabla\log q_t$ (the target to be learned) in score-based diffusion models in Euclidean space [1] is singular as $t\rightarrow 0$ when the learn distribution is defined on a submanifold. For the diffusion models in [2], it is visible from Figure 1 in [2] that the generated samples do not lie exactly on the manifold. While the reviewer is concerned about the added complexity of our algorithm, **we are concerned about the above-mentioned issues (having in mind that these issues exist in high-dimensional applications but can not be visualized)** and we think it is necessary to develop better methods on manifolds.
> >
> > In fact, motivated by the pioneering works [1,2], several groups (including one author of [1]) have recently studied diffusion models on manifolds [3,4,5]. Besides, flow-matching [6] and normalizing flows [7-8] on manifolds have also been developed. Our work aims at making a further contribution to this direction by generalizing the manifolds that can be studied. We do not think that a fair comparison between methods on Euclidean space and methods on manifolds can be easily made, given that in the manifold setting the latter offers methods with sound theoretical basis and studies much more challenging applications, i.e. complex high-dimensional manifolds (In view of the complex manifolds in applications, a simple example like the swissroll dataset alone is not sufficient to guarantee the validity and accuracy of a method).
> >
> >
> > [1] Yang Song and Stefano Ermon. Generative modeling by estimating gradients of the data distribution, NeurIPS, 2019
> >
> > [2] Jascha Sohl-Dickstein, et al. Deep unsupervised learning using nonequilibrium thermodynamics, ICML, 2015
> >
> > [3] Valentin De Bortoli, Emile Mathieu, Michael Hutchinson, James Thornton, Yee Whye Teh, and Arnaud Doucet. Riemannian score-based generative modelling. NeurIPS, 2022
> >
> > [4] Chin-Wei Huang, Milad Aghajohari, Joey Bose, Prakash Panangaden, and Aaron Courville. Riemannian diffusion models. NeurIPS, 2022
> >
> > [5] Aaron Lou, Minkai Xu, Adam Farris, and Stefano Ermon. Scaling Riemannian diffusion models. NeurIPS, 2023
> >
> > [6] Ricky T. Q. Chen and Yaron Lipman. Flow matching on general geometries. ICLR, 2024
> >
> > [7] Heli Ben-Hamu, Samuel Cohen, Joey Bose, Brandon Amos, Maximilian Nickel, Aditya Grover,
> > Ricky T. Q. Chen, and Yaron Lipman. Matching normalizing flows and probability paths on manifolds. ICML, 2022.
> >
> > [8] Emile Mathieu and Maximilian Nickel. Riemannian continuous normalizing flows. NeurIPS, 2020

---

### Official Review · Reviewer_PKgc · 2024-11-03

**Soundness:** 2
**Presentation:** 3
**Contribution:** 3
**Rating:** 5
**Confidence:** 3

**Summary:**

This paper presents a Riemannian Denoising Diffusion Probabilistic Model (RDDPM) designed for learning distributions on submanifolds embedded in a Euclidean space. The key advantage of this approach, compared to existing diffusion models on manifolds, is that it does not require extensive information of the manifold such as its eigenfunctions or geodesics. Instead, it employs a projection scheme that relies solely on the level set function defining the manifold. The authors proposed a training algorithm and analyzed the continuous-time limit of the model. Experimental results demonstrate the effectiveness of the method in learning distributions supported on a known manifold.

**Strengths:**

1. The paper is well-written and well-organized, making it easy to follow.
2. While the idea of using a projection scheme instead of relying on extensive manifold information, such as geodesics and Laplacian eigenfunctions, has been explored in previous works, it remains an interesting approach for defining a noising/denoising Markov chain with (implicitly given) transition kernel.
3. The authors conduct extensive experiments to demonstrate the effectiveness of the proposed model.

**Weaknesses:**

1. One of my main concerns is Equation (8), which defines the transition kernel. Although the map $G_x: \mathcal{M} \to T_x \mathcal{M}$ is well-defined, it is not a true bijection due to the possibility of multiple solutions arising from the constraint in Equation (5). This implies that Equation (8) may not hold in general. While the authors mention in a footnote that $\sigma$ can be chosen small enough to ensure that Equation (5) has at least some solution with high probability, the fundamental issue of multiple solutions cannot be avoided.

2. Unlike Euclidean diffusion models, which do not require explicit forward simulation, the proposed method relies heavily on extensive forward simulation for training, as the transition kernel from time zero to any arbitrary time is not readily available. It would be beneficial to report this additional computational time in the numerical experiments.

3. The equilibrium distribution is generally unknown in the presence of a nontrivial ambient drift $b(x)$. The authors suggest that this distribution can be approximated by running a sufficiently long forward Markov chain. However, this process may be time-consuming, especially since $\sigma$ must be small to address the issue of Equation (5) lacking solutions.

**Questions:**

Please refer to the previous section

---

> ### Author Response · Authors · 2024-11-20
> **Author Response to Reviewer PKgc**
>
> We appreciate the reviewer’s time and effort, as well as the helpful questions and suggestions.
>
> ## - Weaknesses assessed by the reviewer
>
> 1. One of my main concerns is Equation (8), which defines the transition kernel. Although the map $G_x$ is well-defined, it is not a true bijection due to the possibility of multiple solutions arising from the constraint in Equation (5). This implies that Equation (8) may not hold in general. While the authors mention in a footnote that $\sigma$ can be chosen small enough to ensure that Equation (5) has at least some solution with high probability, the fundamental issue of multiple solutions cannot be avoided.
>
>     **Reply**: We thank the reviewer for the careful reading and for raising this question. We agree that there were some issues in our previous analysis leading to Equation (8). In the revised manuscript, we have refined our analysis and those issues are resolved. Please refer to our ``Reply to common questions from reviewers'' above for detailed discussions.
>
> 2. Unlike Euclidean diffusion models, which do not require explicit forward simulation, the proposed method relies heavily on extensive forward simulation for training, as the transition kernel from time zero to any arbitrary time is not readily available. It would be beneficial to report this additional computational time in the numerical experiments.
>
>     **Reply**: In the revised manuscript, we have added the computational cost of the proposed algorithm in Appendix D.1. Please also refer to our ``Reply to common questions from reviewers'' above for discussions on trajectory simulations.
>
> 3. The equilibrium distribution is generally unknown in the presence of a nontrivial ambient drift $b(x)$. The authors suggest that this distribution can be approximated by running a sufficiently long forward Markov chain. However, this process may be time-consuming, especially since $\sigma$ must be small to address the issue of Equation (5) lacking solutions.
>
>     **Reply**: In fact, except for a few very special manifolds, it is generally difficult to directly sample from the prior distribution, even if it is chosen to be the uniform distribution on manifold ($b=0$ in our case). Therefore, one might need to get data distributed according to the prior by sampling certain Markov chains or SDEs on the manifold.
>
>     The length of the trajectory that is needed in order for the data to well approximate the prior depends on the speed of convergence of the Markov chain. In this regard, it may be advantageous to sample a Markov chain with a nonzero drift $b$ if the convergence to equilibrium (a non-uniform prior) is faster compared to the convergence of random walk on manifold to the uniform distribution. This motivates our discussion on nonzero $b$.
>
>     Another observation that is worth mentioning is that the prior data only needs to be sampled once and can be reused afterwards to generate different distributions on the same manifold (in different generative tasks).
>
>     Regarding the step-size $\sigma$, while it should be properly chosen depending on the manifold under consideration, we have refined our analysis so that Equation (5) lacking solution is no longer an issue. Please refer to our ``Reply to common questions from reviewers'' for detailed discussions on this point.

---

> > ### Comment · Reviewer_PKgc · 2024-11-25
> >
> > I thank the authors for their response. However, I share the concerns raised by other reviewers (gqWG and fyyt) regarding the effectiveness and theoretical soundness of the proposed RDDPMs. Specifically, the method does not demonstrate a clear advantage and presents several potential pitfalls, including the choice of sigma, non-uniqueness of projection, and rejected trajectories. For these reasons, my rating remains unchanged.

---

> ### Author Response · Authors · 2024-11-26
>
> We thank the reviewer for the response.
>
> Regarding the advantage, we are comparing our work to prior works on manifolds [1-6] and we have applied our approach to two numerical examples (SO(10) and alanine dipeptide with fixed dihedral angle), both of which have not been studied by existing methods due to their complexity. From these examples, we think the advantage of our method is apparent.
>
> Regarding the non-uniqueness of projection, we would like to emphasize that our analysis concerns a Markov chain where the new state is determined by a selection scheme (numerical solvers such as Newton's method) that picks one solution when multiple solutions exist. The constraint equation may have multiple solutions, but the **non-uniqueness is not an issue to our analysis** (As explained in ``Reply to common questions from reviewers'' and in our revised manuscript, we are working with a bijection.)
>
> Regarding the choice of step-size sigma and rejected trajectories, we were careful at theoretical level in discussing these issues. In practice, the algorithm works for a wide range of sigma. As far as rejected trajectories are concerned, we have reported that, **for the mesh data, there are only less than 1.0% trajectories rejected**. This is mainly due to the existence of the curved regions of the geometry objects and the fact that the level set function is represented as a complex neural network. There is **no rejected trajectories in sphere and SO(10) examples**, and only **one out of $8000$ trajectories in the alanine dipeptide example**.
>
> We would like to point out, as we have mentioned in our manuscript, that the same projection is well studied in Monte Carlo sampling on submanifolds by some leading experts in applied mathematics ([7,8] and chapter 3 of the book [9]). We think that the best way to get a feeling about the algorithm is to test it on concrete examples. For this reason, **we provide our code with detailed instructions and hope they can be helpful to readers**.
>
> [1] Valentin De Bortoli, Emile Mathieu, Michael Hutchinson, James Thornton, Yee Whye Teh, and Arnaud Doucet. Riemannian score-based generative modelling. NeurIPS, 2022
>
> [2] Chin-Wei Huang, Milad Aghajohari, Joey Bose, Prakash Panangaden, and Aaron Courville. Riemannian diffusion models. NeurIPS, 2022
>
> [3] Aaron Lou, Minkai Xu, Adam Farris, and Stefano Ermon. Scaling Riemannian diffusion models. NeurIPS, 2023
>
> [4] Ricky T. Q. Chen and Yaron Lipman. Flow matching on general geometries. ICLR, 2024
>
> [5] Heli Ben-Hamu, Samuel Cohen, Joey Bose, Brandon Amos, Maximilian Nickel, Aditya Grover,
> Ricky T. Q. Chen, and Yaron Lipman. Matching normalizing flows and probability paths on manifolds. ICML, 2022.
>
> [6] Emile Mathieu and Maximilian Nickel. Riemannian continuous normalizing flows. NeurIPS, 2020
>
> [7] Giovanni Ciccotti, Tony Lelièvre, and Eric Vanden-Eijnden. Projection of diffusions on submanifolds: Application to mean force computation. Communications on Pure and Applied Mathematics, 2008.
>
> [8] Emilio Zappa, Miranda Holmes-Cerfon, and Jonathan Goodman. Monte Carlo on manifolds: Sam-
> pling densities and integrating functions. Communications on Pure and Applied Mathematics, 2018.
>
> [9] Tony Lelièvre, Mathias Rousset, and Gabriel Stoltz, Free Energy Computations, A Mathematical Perspective. 2010.

---

### Official Review · Reviewer_3ENY · 2024-11-10

**Soundness:** 4
**Presentation:** 4
**Contribution:** 3
**Rating:** 8
**Confidence:** 3

**Summary:**

Denoising diffusion probabilistic models (DDPM) [1] are a popular class of deep models used for image and video generation, denoising, super-resolution and other applications. These models work in Euclidean space. This manuscript proposes an extension of DDPM from R^n to submanifolds of R^n that are implicitly defined by an equality \xi(x)=0. i.e. the sub-manifold is implicitly defined as a level set of some smooth function \xi(x). The main idea behind the work is to combine the DDPM [1] with submanifold-projected diffusion [2]. The gist of it is that you can add noise to a data-point on the manifold, obtaining a point outside the manifold which is then retracted to the manifold by solving for \xi(x)=0 using Newton iterations. The authors present the exact probability distributions of the forward and backward Markov processes that follow this idea. Describe the method in detail and present several experiments on both "classic" mathematical manifolds such as SO(10) as well as manifolds that approximate meshes (the Stanford bunny and Crane's cow model).

[1] Ho, Jain, Abbeel. "Denoising Diffusion Probabilistic Models". NeurIPS (2020).
[2] Ciccotti, Lelièvre, Vanden-Eijnden. "Projection of diffusions on submanifolds: Application to mean force computation". Communications on Pure and Applied Mathematics. (2007)

**Strengths:**

While I am not an expert on diffusion models, the work appears like a natural extension to the submanifold setting. The paper is easy to read and the experimental section seems thorough enough as it contains several examples from different domains. Furthermore, the accompanied code appears examplary and contains clear instructions on how to reproduce the results.  Overall, this appears to be high-quality work.

**Weaknesses:**

In my opinion there is only one small missing element that is easy to address: the paper should contain more details regarding the computational cost of the proposed procedure. At the very least I would like to see runtimes for the experiments in the paper and a short appendix section explaining all the computational costs associated with the procedure, how they depend on the dimension of the sub-manifold, the complexity of the level-set function, etc. so that readers would have a good idea when and where this method might be applicable.

**Questions:**

* Line 146: So what happens if the solution does not exist? What does your method actually do in that case?
* Line 239: "they should be relatively small" - relative to what?
* Line 529: "We have demonstrated [...] high-dimensional manifolds that can not be easily studies by existing methods" - can you be more specific here about what you demonstrated? Are you referring specifically to the SO(10) example in section 6.3?

A few minor suggestions:
* There are a few places where the language used is a bit odd. For example, just in the abstract we have the following: "most of the manifolds interested in applications" (clearly, manifolds are not interested in anything). "Laplacian-Beltrami" (instead of Laplace-Beltrami), "SO(10) and configuration space" (missing "the").
* Line 081: "gradually destructing" - did you really mean to say destructing here?
* Eq. (5): it is not clear at this point why b(x) is needed as this is only explained later in page 5 so you may want to refer the reader to that explanation here.
* Lines 249-255: an illustration would be nice here showing some submanifold and a helpful choice for b(x).
* Line 320: "DDPMs employ a forward Markov chains" - should be "Markov chain".
* Tables 1,2: it would be better to write "negative log-likelihood" instead of NLL and to also state that "smaller is better" to make the tables clearer to the casual reader who doesn't read the paper but just skims tables and figures.

---

> ### Author Response · Authors · 2024-11-20
> **Author Response to Reviewer 3ENY**
>
> We appreciate the reviewer’s valuable feedback and thoughtful suggestions.
>
> ## -  Weaknesses assessed by the reviewer
>
> 1. In my opinion there is only one small missing element that is easy to address: the paper should contain more details regarding the computational cost of the proposed procedure. At the very least I would like to see runtimes for the experiments in the paper and a short appendix section explaining all the computational costs associated with the procedure, how they depend on the dimension of the sub-manifold, the complexity of the level-set function, etc. so that readers would have a good idea when and where this method might be applicable.
>
>    **Reply**: We thank the reviewer for the positive evaluation and helpful suggestions. We have added a short appendix where we summarize and discuss the computational cost on different examples. Please refer to Appendix D.1 in our revised manuscript for the details regarding the runtime of our experiments and to Appendix D.2 for discussions on the computational complexity of the projection scheme.
>
>
> ## - Questions raised by the reviewer
> 1. Line 146: So what happens if the solution does not exist? What does your method actually do in that case?
>
>     **Reply**: We discuss how the projection is solved in Section 3.5 (Algorithm Details). Specifically, in Line 255, we mention that ``*When no solution is found, one can re-generate the state or the entire trajectory.*" This is also reflected in Lines 7-11 of Algorithm 2 and Lines 6-10 of algorithm 3.
>
>    In our numerical experiment, we have provided details on the percentage of unsuccessful paths. In the second example on mesh data (Standard Bunny and Spot the Cow), only less than $1.0\%$ paths can not be generated. In the fourth example on alanine dipeptide, the case where the solution does not exist happens only once when generating $8000$ trajectories.  In the first example on earth data and the third example on SO(10), the solution can always be found.
>
> 2. Line 239: "they should be relatively small" - relative to what?
>
>    **Reply**: Here we meant that the step-size should be properly chosen depending on the manifold under consideration, e.g. how curved the manifold is. In the revised manuscript, we have rewritten this sentence to clarify this point.
>
> 3. Line 529: "We have demonstrated [...] high-dimensional manifolds that can not be easily studies by existing methods" - can you be more specific here about what you demonstrated? Are you referring specifically to the SO(10) example in section 6.3?
>
>    **Reply**: Here we mean the SO(10) and alanine dipeptide examples, which can not be handled by existing methods that rely on knowing heat kernel or eigenfunctions of the Laplace-Beltrami operator of the manifold. Following the reviewer's suggestion, we have specified what we mean in the following sentence before we conclude the Introduction.
>
>    > ``*We successfully apply our method to datasets from prior works, and to new datasets from the special orthogonal group $\mathrm{SO}(10)$ and from alanine dipeptide with fixed dihedral angle, both of which are challenging for existing methods due to their geometric complexity.*"
>
> 4. A few minor suggestions...
>
>    **Reply**: We thank the reviewer for the comments. We have gone through our manuscript thoroughly and polished the writing again.

---

### Author Response · Authors · 2024-11-20
**Reply to common questions from reviewers**

Dear reviewers,

  We would like to thank you all for your thoughtful comments on our work. We have carefully revised our manuscript based on your suggestions. Our reply to each reviewer's questions can be found below the corresponding official review. Here, we would like to clarify a few points in our paper that are concerned by multiple reviewers. Please refer to our revised manuscript for concrete changes. We are happy to answer your further questions.

## 1. Derivation of the transition density of Markov chain

In our initial submission, we stated in Section 3.1 that

> ``*We assume that $\sigma$ is small, such that the case where the solution to equation 5 does not exist is negligible.*''

and in footnote 1, we said:

> ``*We point out that in deriving equation 8 we have omitted the case where there may be no solution or multiple solutions to the constraint equation 5 for some vectors $v$. This is valid as long as $\sigma$ is reasonably
small. ..*''

During the revision, we re-examined our analysis and realized that the discussions quoted above are not precise. In the following, we explain how **we have refined our analysis so that now it takes into account the cases where the constraint equation has multiple solutions or no solution**.

We derive the transition density of the Markov chain by a change of variables for probability densities. For this purpose, we need to verify that the map (denoted by $f$) from a tangent vector $v$ to the projected state $y$ is a bijection. While the constraint equation (5) may have multiple solutions, we are considering a Markov chain where the new state is determined by a selection scheme that picks one solution when multiple solutions exist. As discussed in our paper, such a selection scheme can be instantiated using Newton’s method with fixed initial condition. The map $f$ defined by this selection scheme is apparently one-to-one, and we are deriving the transition density under this selection scheme. This resolves the issue due to multiple solutions.

 Concerning the issue due to nonexistence of solutions, we explicitly define the domain of $f$ to be the set $\mathcal F_{x,\sigma}$ consisting of all tangent vectors $v$ for which the solution to the constraint equation can be found. In this way, $f$ is a bijection from $\mathcal F_{x,\sigma}$ to its image and we obtain the transition density in equation (8) by applying the change of variables for densities. An extra term appears in the expression of the transition density because the distribution of $v$ in $\mathcal F_{x,\sigma}$ is Gaussian with a rescaled normalizing constant (such that the integral over $\mathcal F_{x,\sigma}$ equals one). This leads to an extra term in  the transition densities of both forward and reverse processes, but the final training objective remains unchanged.

Here, we keep the discussion brief and further details can be found in Section 3.1 in our revised manuscript (text in blue).

## 2. Choice of step-size $\sigma$

With our refined analysis, a large step-size is no longer an issue for the theoretical derivation.  In practice, we suggest that it should be chosen properly (depending on the concrete application), so that on the one hand the total number of steps required to reach equilibrium is not too large and on the other hand a new state can be found by the Newton's method with high probability and the trajectory simulation is efficient. A potential advantage of small step-size is that the Newton's method requires fewer iterations (faster) because it starts from a good initial guess.

## 3. Computational cost for trajectory simulation

Unlike the Euclidean setting, existing methods on manifolds (e.g. score-based diffusion models and flow-matching on manifolds) need to simulate trajectories, e.g. geodesic random walk or flows, except for a few special manifolds. Our method is the same in this respect.

In Section 3.5, we mention that generating trajectories can be done in a preparatory
step before training, and the trajectory dataset can be updated every several
training epochs. We have reported the computational cost of trajectory simulation in the revised manuscript. Except for the earth data where the overall training is fast, the percentage of simulation time is generally below $12.0\%$ of the total runtime. Please refer to Tables 4-5 in Appendix D for details.

---

### Meta-Review · Area_Chair_KChD · 2024-12-20

**Metareview:**

The paper proposes Riemannian Denoising Diffusion Probabilistic Models (RDDPMs) for learning distributions on submanifolds of Euclidean space that are level sets of functions. In constrast to existing methods for generative modeling on manifolds that rely on geometric information such as geodesic curves or eigenfunctions of the Laplace-Beltrami operator, the proposed method is built on a projection scheme and requires being able to evaluate the value and the first order derivatives of the function that defines the submanifold.  Theoretical analysis in the continuous-time limit and numerical experiments are provided to illustrate the method.

Reviewers generally appreciate the proposed projection scheme and the extensive experiments carried out in the paper. However, there are many concerns regarding the soundness of the proposed method. One major concern, among others, is the issue of potentially no solution or multiple solutions to Eq(5) (Reviewers  PKgc,  gqWG,  fyyt) in which latter case the projection is not unique.

Due to these concerns over the soundness of the method,  I cannot recommend acceptance of the manuscript in its current form.

**Additional Comments On Reviewer Discussion:**

In their rebuttal, the authors revised the manuscript and argued that the probability of no solution goes to $0$ as $\sigma \rightarrow 0$. Furthermore, it is assume that in the case of multiple solutions, a numerical solver finds one solution in a deterministic way. When no solution is found, the authors suggested to re-generate the state or the entire trajectory though this does not guarantee a resolution of this issue.The reviewers thus remained  unconvinced about the benefits of the proposed method given these potential pitfalls.

---

### Decision · Program_Chairs · 2025-01-22

Reject